# Sensitivity analysis of wake steering optimisation for wind farm power maximisation

Filippo Gori , Sylvain Laizet , and Andrew Wynn

Department of Aeronautics, Imperial College London, SW7 2AZ London, UK

**Correspondence:** Filippo Gori (f.gori21@imperial.ac.uk)

**Abstract.** Modern large–scale wind farms consist of multiple turbines clustered together, usually in well–structured formations. Clustering has a number of drawbacks during a wind farm's operation, as some of the downstream turbines will inevitably operate in the wake of those upstream, with a significant reduction in power output and an increase in fatigue loads. Wake steering, a control strategy in which upstream wind turbines are misaligned with the wind to redirect their wakes away from downstream turbines, is a promising strategy to mitigate power losses. The purpose of this work is to investigate the sensitivity of open-loop wake steering optimisation in which an internal predictive wake model is used to determine the farm power output as a function of the turbine yaw angles. Three different layouts are investigated with increasing levels of complexity. A simple $2 \times 1$ farm layout in aligned conditions is first considered, allowing for a careful investigation of the sensitivity to wake models and operating conditions. A medium-complexity case of a generic $5 \times 5$ farm layout in aligned conditions is examined, to enable the study of a more complex design space. The final layout investigated is the Horns Rev wind farm (80 turbines), for which there have been very little studies of the performance or sensitivity of wake steering optimisation. Overall, the results indicate a strong sensitivity of wake steering strategies to both the analytical wake model choice, and to the particular implementation of algorithms used for optimisation. Significant variability can be observed in both farm power improvement and optimal yaw settings, depending on the optimisation set-up. Through a statistical analysis of the impact of optimiser initialisation and a study of the multi-modal and discontinuous nature of the underlying farm power objective functions, this study shows that the uncovered sensitivities represent a fundamental challenge to robustly identifying globally optimal solutions for the high-dimensional optimisation problems arising from realistic wind farm layouts. This paper proposes a simple strategy for sensitivity mitigation by introducing additional optimisation constraints, leading to higher farm power improvements and more consistent, coherent, and practicable optimal yaw angle settings.

## 1 Introduction

Wind energy now plays a central role in meeting world energy requirements, driven by the urgent need to mitigate climate change and a significant recent reduction in its levelised cost. New and ambitious international renewables targets, such as the European Commission's proposed installation of up to $450$ GW offshore wind capacity by 2050 (Pryor et al., 2021), will require further significant increases in wind energy capacity. Large wind arrays with closely spaced turbines will be used to meet this demand. These have logistical advantages over sparse layouts in terms of lower infrastructure and maintenance costs

and increased energy density. However, the performance of large wind arrays is degraded by the aerodynamic interactions between turbines. Upstream-generated turbine wakes exhibit lower wind speeds and increased turbulence intensity, causing typical annual power losses of 10 % to 30 % (Nygaard, 2014; Barthelmie et al., 2009, 2010) in addition to increased fatigue loading.

Wake steering, in which upstream turbines are yawed to deflect their wakes away from downstream machines, is a leading control technique used to mitigate the effects of wake-turbine interactions. Farm power increases ranging from 5 % to 47 % have been demonstrated in high-fidelity simulations (Fleming et al., 2015, 2018), wind tunnel campaigns (Campagnolo et al., 2016; Bastankhah and Porté-Agel, 2019; Zong and Porté-Agel, 2021), and field tests (Howland et al., 2019). To successfully implement wake steering, one must navigate a natural trade-off: while wake steering can increase wind speed and power
production at downstream turbines, it also reduces both the effective swept area and power production of any yawed wake-generating turbine. For large arrays of turbines, in which there are multiple competing trade-offs arising from each interacting pair of turbines, the question of maximising farm power via wake steering becomes both a high-dimensional and non-convex optimisation problem. Such problems can be challenging to solve, and their solutions can exhibit high parametric sensitivity, which may partially explain the wide range of reported farm power improvements in the literature.

With a view to enabling robust and predictable wake steering optimisation for large wind farms, this paper investigates the *sensitivity* of wake steering optimisation. We focus on open-loop approaches in which internal predictive wake models are used to determine the farm power output (the cost function to be maximised) as a function of the turbine yaw angles (the decision variables). In this paper, we will use analytical, steady-state, wake models which capture only the most dominant physical wake features. A number of models will be considered, including the Jensen (Jensen, 1983), Multizone (Gebraad et al., 2014),
Gaussian (Bastankhah and Porté-Agel, 2014) and Gauss-Curl-Hybrid (King et al., 2021) models, each of which is suitably adapted using the approaches of Jiménez et al. (2009) and Bastankhah et al. (2022) to capture yaw-induced wake modifications. These are chosen to be representative of common models used in wake-steering optimisation (Kheirabadi and Nagamune, 2019). Many optimisation algorithms have been used to implement wake steering, and these can be broadly categorised as either gradient-based or gradient-free algorithms. Gradient-based methods include gradient ascent (Zong and Porté-Agel, 2021),
sequential quadratic programming (Annoni et al., 2018), steepest descent (Howland et al., 2019), conjugate gradient (Thøgersen et al., 2017), and Quasi-Newton methods (van Dijk et al., 2017). Gradient-free approaches to open-loop wake steering studies have used game theory (Gebraad et al., 2016; van Dijk et al., 2016; Rott et al., 2018), statistical optimisation for dynamic wind directions (Simley et al., 2020), particle swarm optimisation (Ahmad et al., 2019; Dou et al., 2020), evolutionary algorithms (Dou et al., 2020), and random search algorithms (Kuo et al., 2020). With a wide choice of models and algorithms available, it
is natural to question if the solution to a given wake steering optimisation problem is sensitive to these choices.

     By optimisation sensitivity, we refer to the dependency of wake steering optimisation on (i) the choice of the underlying predictive model; (ii) the choice of optimisation algorithm and its particular parametric implementation; and (iii) the given operating condition, such as farm layout or atmospheric conditions. Furthermore, we measure optimisation sensitivity in terms of both (i) the optimised farm power; and (ii) the optimal decision variables themselves, i.e., the optimal yaw angles obtained
from wake steering optimisation. Arguably, the latter is the most important. Of particular interest is to identify situations of high

sensitivity: that is, when either the predicted maximised power output, or the optimal yaw angles, are seen to vary substantially with small changes to the wake model, algorithm, or operating condition. When high sensitivity is identified, we will also study potential algorithmic changes to reduce it.

Understanding, quantifying, and reducing cases of high sensitivity is of fundamental importance for robust wake steering. To implement open-loop wake steering, tuning or identification of wake model parameters from field data or high-fidelity simulations is required. As with any tuned model, there can be a mismatch between the predicted and true farm behaviour. A first problem arises when the predicted optimised farm power output is highly sensitive to the wake model parameters or algorithmic implementation. In these cases, small modelling mismatches may cause large deviations between expected and true farm performance, possibly nullifying any predicted power increase when applied in the field. A second problem occurs in the case of high optimal yaw angle sensitivity. Here, a small parametric change (e.g., to model re-tuning, or to atmospheric conditions) may require large yaw angle changes, and it is practically undesirable to make large control input changes in response to only small operational perturbations. As will be discussed in Sect. 4.3, the sensitivity of optimal yaw angles can be observed in recent open-loop wake steering studies for large-scale farms such as the Horns Rev installation (Dou et al., 2020). More generally, we will show that the high sensitivity of decision variables also presents a fundamental challenge which must be overcome when one seeks to globally optimise wake steering strategies for the high-dimensional optimisation problems arising from realistic farm layouts.

In this paper, we do not consider active wake control or closed-loop wake control in which online sensor information is used to dynamically adapt wake steering strategies (see Kheirabadi and Nagamune (2019) and Houck (2021) for comprehensive reviews). While recent and promising feedback control strategies have been proposed (Doekemeijer et al., 2019, 2020; Howland et al., 2020, 2022), closed-loop approaches still typically employ predictive wake models in order to calculate their time-varying control inputs. Consequently, the question of predictive sensitivity is still important in the more complex closed-loop setting. By first studying open-loop wake optimisation strategies in this paper, we will therefore also provide insight into the behaviour of closed-loop approaches to wake steering optimisation.

Due to the large number of underlying modelling, algorithmic, and parametric choices, in this paper we propose a hierarchy of test-cases to facilitate the understanding of the different sensitivities present in wake steering optimisation. At the simplest level, a minimal optimisation problem consisting of a $2 \times 1$ farm layout in aligned conditions is first considered. In this case, optimisation can be performed by a simple parametric sweep, removing any sensitivity to algorithmic choice or implementation. The $2 \times 1$ case, therefore, allows a careful investigation of sensitivity to only the underlying wake model and operating conditions. Next, a medium-complexity case of a $5 \times 5$ farm layout in aligned conditions is considered. This may be viewed as several non-trivially interacting $2 \times 1$ blocks, leading to a more complex design space. Sensitivity, therefore, depends on both the algorithmic implementation and underlying wake model, but quantification of the $2 \times 1$ case facilitates the decoupling of these two dependencies. We present a statistical analysis of optimiser sensitivity in this case, in addition to comparing the performance of global stochastic and local gradient-based optimisation algorithms. Finally, a representative layout corresponding to the Horns Rev wind farm is considered. The investigation of this more complex optimisation problem will be used to confirm whether the wake steering sensitivities uncovered in the low and medium complexity cases transfer to a realistic setup. Finally,

we propose a simple strategy for sensitivity mitigation, based on appropriately chosen optimisation constraints, which is effective even for this realistic and complex wake steering optimisation problem. To the best of our knowledge, previous parametric investigations into wake optimisation sensitivity (Rak and Santos Pereira, 2022; Göçmen et al., 2022) have not studied the effect of optimiser class or analysed sensitivity from a statistical perspective, and have not presented possible strategies for sensitivity mitigation.

The remainder of the paper is organised as follows. Section 2 gives an overview of the wind farm modelling framework and introduces the wake and deflection models. In Sect. 3, a description of the optimisation setup is provided, including wind farm layouts, atmospheric conditions, and optimisation algorithms. Sensitivity results and discussion are presented in Sect. 4 for the $2 \times 1$ (Sect. 4.1), $5 \times 5$ (Sect. 4.2), and Horns Rev (Sect. 4.3) cases. Finally, conclusions are given in Sect. 5. Appendices provide additional details on wake and deflection model formulations (Appendix-A), a validation of the optimisation framework (Appendix-B), and an illustrative example of a Bayesian optimisation on a 1-D toy problem (Appendix-C).

## 2   Wind farm modelling

Wind farm modelling in the present work is conducted with version 2.4 of the open-source `FLORIS` framework (NREL, 2021) by the National Renewable Energy Laboratory (NREL). The wake models considered in this paper are Jensen (Jensen, 1983), Multizone (Gebraad et al., 2014), Gaussian (Bastankhah and Porté-Agel, 2014), and Gauss-Curl Hybrid (GCH) (King et al., 2021) and their respective deflection models (Jiménez et al., 2009; Bastankhah et al., 2022). The four selected models, fully described in Appendix-A, have different complexities, physical modelling capabilities, and empirical parameter dependencies. They are selected because they are representative of the models commonly used by the wind energy community in optimisation and control studies (Kheirabadi and Nagamune, 2019).

Although different wake models will logically give different power predictions, in a wake steering optimisation context, it may still be the case that different models give rise to similar optimal yaw angles. Consequently, the question of the modelling fidelity required to enable robust wake steering optimisation may be different to the question of which model is best to capture a given physical wake characteristic. In this regard, we emphasise that the purpose of this study is not to identify which model is best to study a particular wind farm. Rather, we seek to identify situations in which wake steering optimisation is highly sensitive either to the choice of underlying model, or to the interaction between the model choice and the particular parametric implementation of commonly-used optimisers (see Sect. 3 for a full description of optimisation algorithms used), since high sensitivity presents a fundamental barrier to robust optimisation performance, even using models tuned to high-fidelity simulation data or field data. As a result, the wake and deflection model parameters for this study simply use the `FLORIS` recommended standard values, reported in Table 1.

We now give a brief overview of `FLORIS`'s wind farm modelling structure. First, an initial condition is defined by specifying atmospheric inflow, wind farm layout, turbine geometry, and operational conditions. Next, the chosen wake model calculates each turbine's steady streamwise velocity deficit. The computation is sequential to allow additional considerations on added turbulence. For yawed turbine cases, a deflection model is employed to determine and apply a cross-stream shift in the stream-

wise velocity deficit field. Finally, streamwise velocity deficits for each turbine are combined with a superposition model and are applied to the initial flow field. In the current study, all wake models use the sum of squares freestream superposition (SOSFS) model developed by Katic et al. (1987). Based on the conservation of the mean kinetic energy deficit during wake interaction, the combined wake velocity $U_w$, dependant on the three spatial dimensions $(x, y, z)$, is defined as

$$U_w(x, y, z) = U_\infty - \sqrt{\sum_i \left(U_\infty - u_w^i(x, y, z)\right)^2} \,, \tag{1}$$

where $U_\infty$ is the mean free stream velocity, and $u_w^i$ is the wake streamwise velocity induced by turbine $i$ in stand-alone conditions.

Farm power calculations are conducted as follows. Turbine operational profiles consist of lookup tables for power and thrust coefficients ($C_P$ and $C_T$, respectively) as a function of streamwise velocity. These tables are generated by the FAST (Jonkman, 2021) (Fatigue, Aerodynamics, Structures, and Turbulence) aeroelastic simulator developed by NREL. Given a resolved wind farm flow field $U_w(x, y, z)$, the streamwise velocities at the turbine rotor grid points $U_{\text{rotor}}(x, y, z)$ are defined by interpolation. The averaged streamwise velocity $\overline{U}$ of the spatial extent of the un-yawed turbine is expressed by

$$\overline{U} = \sqrt[3]{\overline{U_{\text{rotor}}^3}} \,. \tag{2}$$

The power $P_T$ generated by a single turbine is given by

$$P_T = 0.5 \rho A C_P (\overline{U}(\cos \gamma)^w)^3 \,, \tag{3}$$

where $\rho$ is the air density, $A$ is the un-yawed rotor swept area, $C_P$ is the interpolated power coefficient based on the rotor averaged streamwise velocity $\overline{U}$, $\gamma$ is the turbine yaw angle to the inflow wind direction, and $w$ is a tuneable parameter matching the power loss due to yaw misalignment. This study uses $w = 0.627$ for both the turbines considered (NREL 5 MW and Vestas V-80 2 MW), matching the tuned value for the NREL 5 MW presented in Fleming et al. (2017). Considering the lack of power data for the Vestas V-80 2 MW turbine in yawed conditions, preliminary investigations were conducted to assess the influence of $w$ on yaw steering optimisation. It was found that $w$ does not fundamentally impact the main trends of optimal yaw settings and power improvements presented herein, motivating the use of a common value $w = 0.627$ for all examples. The total farm power is calculated as

$$P_{WF} = \sum_{i=1}^{N} P_{Ti}, \tag{4}$$

where $N$ is the total number of turbines in the farm configuration. Finally, the normalized farm power production is defined as

$$G(\boldsymbol{\gamma}) = \frac{P_{WF}(\boldsymbol{\gamma})}{P_{WF}(\mathbf{0})}, \tag{5}$$

where $\boldsymbol{\gamma} = \{\gamma_i\}_{i=1}^{N} \in \mathbb{R}^N$ are the yaw angles of the $N$ turbines considered in each configuration, $P_{WF}(\mathbf{0})$ is the farm power without wake steering, and $P_{WF}(\boldsymbol{\gamma})$ is the farm power achieved for a particular choice of yaw angles $\boldsymbol{\gamma}$.

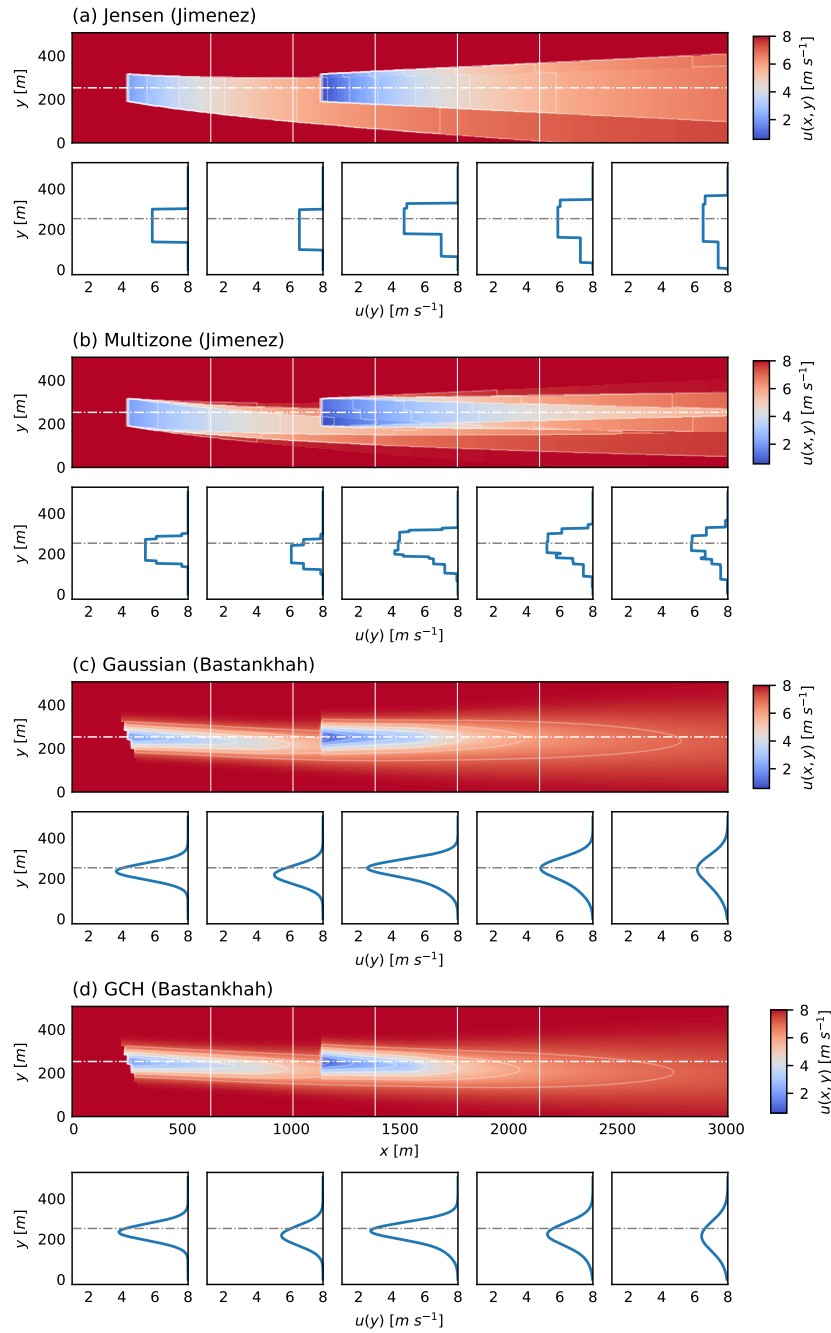

**Figure 1.** Streamwise velocity at hub height of a $2 \times 1$ wind farm layout for the (**a**) Jensen (Jimenez), (**b**) Multizone (Jimenez), (**c**) Gaussian (Bastankhah) and (**d**) GCH (Bastankhah) wake and deflection models at an upstream rotor yaw angle of $20°$. White vertical lines represent the streamwise locations for the velocity profile section plots, ordered by increasing downstream distance, while the horizontal lines indicate the upstream rotor centreline.

Figure 1 presents a visual overview of the main wake features predicted by the models employed in this study, including streamwise velocity deficit distributions, expansion rates, and wake deflections due to yaw. This figure will be referred to subsequently to help understand the sensitivity of wind farm wake steering optimisation to the choice of underlying wake models. Further details on wake and deflection model formulations are provided in Appendix-A.

## 3 Optimisation and wind farm setups

Open-loop wake steering optimisation for power maximisation is conducted on three different farm layouts. The first two are a $2 \times 1$ farm layout and a $5 \times 5$ farm layout, both in fully aligned wind conditions, with NREL 5 MW turbines (Jonkman et al., 2009) with rotor diameter $D = 126$ m and hub height $H = 90$ m. In the base condition, turbine spacing is set to 7 rotor diameters in the streamwise direction and 5 rotor diameters in the cross-stream direction. The third farm layout considered is representative of the offshore Horns Rev wind farm (Hansen et al., 2012). It consists of 80 Vestas V-80 2 MW turbines with rotor diameter $D = 80$ m and hub height $H = 70$ m, arranged in a rhomboid shape. For the considered wind direction ($wd = 270\,^\circ$), the farm exhibits an aligned layout with ten rows and eight columns and a distance between two consecutive turbines of $7\,D$ for both the streamwise and cross-stream directions.

For all the optimisation test-cases, the incoming flow is fully aligned with the farm columns with a wind speed of $8\ \mathrm{m\ s^{-1}}$ in $5\,\%$ ambient turbulence intensity. Wind shear and veer are not taken into account. The sensitivity of wake steering strategies to operating conditions is only investigated for the minimal $2 \times 1$ case, where variations in streamwise spacing and ambient turbulence intensity are performed. Validation for the employed optimisation framework is conducted by comparing results with the findings by Rak and Santos Pereira (2022) and is provided in Appendix-B.

The optimisation problem's objective function in all cases is chosen to be the normalised farm power production (Eq. 5). In all cases, we solve the constrained maximisation problem

$$\max_{\boldsymbol{\gamma}} \quad G(\boldsymbol{\gamma})$$

$$\text{subject to} \quad -25\,^\circ \le \gamma_i \le 25\,^\circ, \quad 1 \le i \le N,$$

(6)

which imposes typical operational bounds of $\pm 25\,^\circ$ on the yaw misalignment of upstream turbines. In Sect. 4.3, additional constraints corresponding to modified bounds and yaw angle monotonicity will be added to the optimisation problem (6) to mitigate optimisation sensitivity. These constraints are imposed by adding further linear inequality constraints to (6) and consequently do not result in any intrinsic increase in complexity of the optimisation problem.

### 3.1 Optimisation algorithms and parameter settings

The optimisation algorithms used are the Sequential Least SQuares Programming (SLSQP) method developed by Kraft (1988) and the Trust Region Bayesian Optimisation (TuRBO) approach by Eriksson et al. (2019). The gradient-based optimiser SLSQP is the standard algorithm used in the `FLORIS` framework. In contrast, TuRBO uses a probabilistic approach for the stochastic global optimisation of large-scale high-dimensional problems, and we now give a brief overview of this alternative algorithm.

The first step is to take an initial sample of design space via Latin Hypercube Sampling (LHS). This generates near-random samples of the objective function from a multi-dimensional distribution. Two subsequent stages then follow. First, the available samples are fitted with a Gaussian process, a probabilistic surrogate model, to give a posterior distribution which both approximates the objective function and determines the approximation's uncertainty. Second, an acquisition function, Thompson Sampling (TS) in the case of TuRBO, is minimised to determine the next sample point at which to evaluate the objective function. The role of the acquisition function is to find a balance between minimising the objective function and reducing the uncertainty of the fitted surrogate model. The two optimisation steps are iterated until pre-defined stopping criteria are met.

The global statistical nature of Bayesian optimisation has been found to outperform local, gradient-based, algorithms when applied to multi-modal or discontinuous objective functions (Shahriari et al., 2016; Eriksson et al., 2019). At each iteration, TuRBO's global approach can potentially sample from anywhere in the design space, while gradient-based SLSQP typically takes a small step computed using local gradients. A known drawback of Bayesian optimisation is the higher computational complexity involved in high-dimensional problems which require a large number of objective function evaluations. To address this issue, TuRBO uses a trust region approach in which multiple Gaussian process surrogate models are fitted in evolving promising areas of the design space, referred to as trust regions. The allocation of new samples to evaluate across the models is achieved via an implicit bandit approach. Further explanation of Bayesian optimisation is given via an illustrative example in Appendix-C.

In this study, the number of initial evaluations generated by the LHS method is set to double the number of optimisation variables $N$ (here the number of turbines). Moreover, the number of trust regions is set to one. To explain this choice, note that if multiple trust regions are used, TuRBO will perform a multiple starting point search and fit a Gaussian process in each trust region, thus conducting multiple sub-searches in parallel. A single TuRBO optimisation run would then be comparable, in complexity, to multiple SLSQP ones. To ensure a fair comparison between the two methods, a single TuRBO trust region is used, initialised with the same conditions as those for each SLSQP optimisation. The parameters used for SLSQP and TuRBO optimisers are shown in Table 2.

As will be discussed subsequently for the $5 \times 5$ case (see, e.g., Fig. 6), the considered objective functions $G(\boldsymbol{\gamma})$ typically exhibit multi-modality, flat regions and discontinuities. Multi-modality refers to the presence of multiple and distinct maxima. These features present a fundamental challenge when seeking to robustly identify global maxima. For the TuRBO algorithm, preliminary investigations indicated only a low parametric influence on optimisation results, with the recommended parameter values in Eriksson et al. (2019) identified as the most sensible choice. These findings are consistent with the stochastic nature of this global optimiser. In the case of SLSQP, a sensible choice of optimisation parameters is required to address this complexity and to essentially regularise the optimisation problem. The parameter to consider is the precision goal for the value of the objective function in the stopping criteria, called $ftol$. It corresponds to the achievement of equal objective function values in the evaluations performed for the numerical approximation of the gradient. Large values of $ftol$ may cause the optimiser to terminate in flat regions, significantly increasing the variability of the objective function and the optimal decision variables to the choice of optimisation initialisation. However, a small $ftol$ typically increases the number of objective function evaluations required for convergence.

In this study, $ftol = $1e-16 for the $5 \times 5$ case is identified as a suitable trade-off between these two behaviours, based on preliminary investigations. For the Horns Rev case, the value $ftol = $1e-12 is used due to the higher dimensionality of the optimisation problem, i.e., its larger computational requirements.

**Table 1.** Wake and deflection model parameters.

| Model | Parameter | Parameter value |
|---|---|---|
| Jensen | $k$ (on/offshore) | 0.05 or 0.04 |
| (Jimenez) | $k_d$ | 0.05 |
| Multizone | $k_e$ | 0.05 |
| (Jimenez) | $m_{e,q}$ | [-0.5, 0.3, 1.0] |
| | $M_{u,q}$ | [0.5, 1.0, 5.5] |
| | $a_u$ | 12.00 |
| | $b_u$ | 1.30 |
| | $k_d$ | 0.05 |
| Gaussian - GCH | $k_a$ | 0.380 |
| (Bastankhah) | $k_b$ | 0.004 |
| | $\alpha$ | 0.580 |
| | $\beta$ | 0.077 |
| | $\epsilon$ | 0.2D |
| | $\phi$ | 2.000 |

**Table 2.** SLSQP and TuRBO parameters.

| Optimiser | Parameter | Parameter value |
|---|---|---|
| SLSQP ($5 \times 5$) | $ftol$ | 1e-16 |
| | $eps$ $[-25\,°, +25\,°]$ | 5e-2 |
| SLSQP (Horns Rev) | $ftol$ | 1e-12 |
| | $eps$ $[-25\,°, +25\,°]$ | 5e-2 |
| | $eps$ $[0\,°, +25\,°]$ ($\rightarrow$ Constraint C1) | 1e-1 |
| TuRBO | $\tau_{\text{succ}}$ | 3.0 |
| | $\tau_{\text{fail}}$ | 25.0 |
| | $L_{\min}$ | 7.8125e-3 |
| | $L_{\max}$ | 1.6 |
| | $L_{\text{init}}$ | 0.8 |

The second SLSQP parameter to consider is $eps \in [0,1]$, corresponding to the step-size (in this case equal to $eps \times [25\,° - (-25\,°)]$) used for the finite difference gradient approximation. Large $eps$ values lead to inaccurate gradient approximations. However, as shown in Fig. 6, the objective functions considered in this study may have either discontinuities or discontinuous derivatives, especially when using Jensen and Multizone models. This 'rough' objective function behaviour, in conjunction with small $eps$ values, may cause the SLSQP optimiser to identify optimal decision variables in local maxima. A $2.5\,°$ step-
size ($eps = 0.05$) is identified to be a suitable value since this relatively large value of $eps$ has the effect of artificially smoothing the rough objective function while limiting the loss of information about the local gradient.

## 4 Results

### 4.1 Two-turbine case - NREL 5 MW

We consider the problem of farm power maximisation for a $2 \times 1$ wind farm layout with NREL 5 MW turbines. The only
235 optimisation variable is the yaw angle $\gamma_1 \in [-25\,°, 25\,°]$ of the upstream turbine. The optimal yaw angle of the furthest downstream turbine is necessarily zero since no other turbine would benefit from a deflection in its wake, and a non-zero

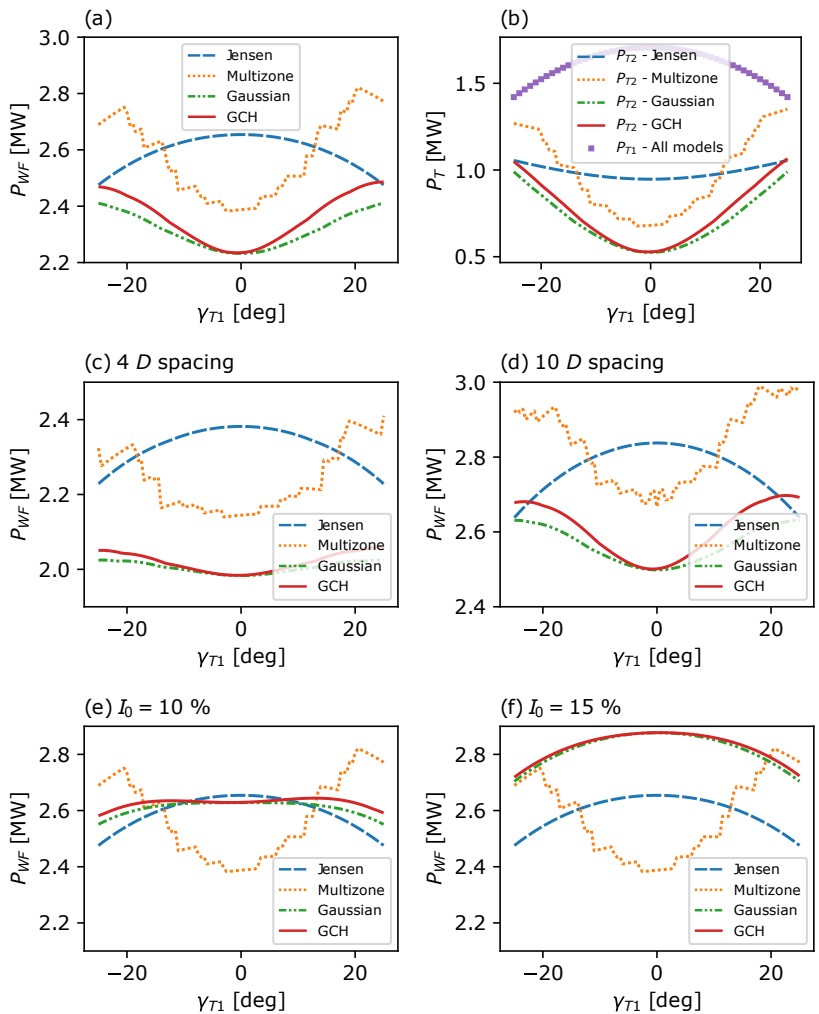

**Figure 2.** Power function of a $2 \times 1$ wind farm layout as upstream turbine yaw angle $\gamma_1$ ranges $\pm 25\,°$. Top plots: wind farm power $P_{WF}$ (**a**), upstream turbine power $P_{T_1}$ (**b**), and downstream turbine power $P_{T_2}$ (**b**) for $7\,D$ streamwise turbine spacing and ambient turbulence intensity $I_0 = 5\,\%$. Middle plots: wind farm power $P_{WF}$ in $I_0 = 5\,\%$ with variable streamwise spacing: $4\,D$ (**c**) and $10\,D$ (**d**). Bottom plots: wind farm power $P_{WF}$ for $7\,D$ downstream spacing and variable turbulence intensity: $I_0 = 10\,\%$ (**e**) and $I_0 = 15\,\%$ (**f**).

yaw angle reduces the power output of this turbine. Optimisation is therefore performed via a simple parameter sweep of $\gamma_1$. Different streamwise turbine spacings and ambient turbulence intensity are considered to provide insights into the sensitivity of both optimised farm power and the optimal yaw angle, denoted $\gamma_1^*$, to operational conditions. Observations made for this minimal study will be used to explain and analyse the sensitivity and performance of wake steering optimisation for more complex farm layouts in Sect. 4.2 and 4.3.

Figures 2-a,b show the extracted power $P_{WF}(\gamma_1)$ of the entire $2 \times 1$ farm, $P_{T_1}(\gamma_1)$ of the upstream turbine, and $P_{T_2}(\gamma_1)$ of the downstream turbine. Power production $P_{T_1}(\gamma_1)$ of the upstream machine is the same for all models. A clear sensitivity to model choice is observed in Fig. 2-a with the Jensen model's optimal yaw angle at $\gamma_1^* = 0°$ while, in contrast, all other models have optimal yaw angles close to constraint boundaries with $\gamma_1^* > 20°$. Model sensitivity is caused by the flatness of the Jensen model's downstream power curve $P_{T_2}(\gamma_1)$ apparent in Fig. 2-b, that is, $\frac{d}{d\gamma_1}P_{T_2}(\gamma_1)$ is significantly smaller for Jensen than for other models. Flatness arises due to the uniform, or "top-hat", profile of the Jensen distribution (see Fig. 1), which results in a lack of sensitivity of streamwise velocity deficit to moderate yaw perturbations. In contrast, it can also be seen in Fig. 1 that all other models predict a streamwise velocity deficit which is both larger at the centreline and decreases more rapidly in the transverse direction. For the Multizone model, this decrease in streamwise velocity deficit is discontinuous due to the model's definition of splitting the wake into various zones, with sharp interfaces between each zone. Overall, Gaussian, GCH and Multizone models give downstream power curves $P_{T_2}(\gamma_1)$ which are more sensitive to changes in $\gamma_1$ than the Jensen model.

Model sensitivity is influenced by both turbine spacing and turbulence intensity $I_0$. Figures 2-c,d show that the optimal yaw angle for the Gaussian, GCH and Multizone models decreases with increased turbine spacing. Although not shown, $\gamma_1^* \approx 0°$ for all models for spacing greater than $32 D$. Similar behaviour can be observed in Fig. 2-e,f as $I_0$ increases, with $\gamma_1^* \approx 0°$ whenever $I_0 \geq 15\%$ for all models which incorporate this parameter (the Multizone model does not depend on $I_0$). From an optimisation perspective, increased spacing and turbulence intensity have the analogous effect of increasing wake recovery and effective wake diameter at the downstream turbine. This reduces the sensitivity of $P_{T_2}(\gamma_1)$ to $\gamma_1$ and renders the wake steering optimisation problem trivial. Despite this observation, it is clear from Fig 2 that for any practically-relevant turbine spacing of up to 10D, and for turbulence intensities of $I_0 \leq 10\%$ (at a typical $7 D$ spacing), it is still the case that substantial model sensitivity is present in the $2 \times 1$ farm power maximisation problem.

With a view towards understanding the sensitivity of wake steering optimisation for more complex farms, it is important to highlight the *multi-modality* of the farm power curves $P_{WF}(\gamma_1)$ for the Gaussian, GCH and Multizone models. Multi-modality can be seen for all considered spacings in Fig. 2-a,c,d, and for $I_0 \leq 10\%$ in Fig. 2-a,e. For the Gaussian model, this arises simply due to the symmetry of $P_{T_2}(\gamma_1)$ for the aligned inflow conditions considered here. Multizone and GCH models are weakly asymmetric due to the incorporation of wake rotation effects, with $P_{T_2}(\gamma_1) > P_{T_2}(-\gamma_1)$, for $\gamma_1 > 0$, in the case considered here of a clockwise spinning rotor. However, the farm power $P_{WF}(\gamma_1)$ curves for both models are still clearly multi-modal.

In summary, this minimal $2 \times 1$ case demonstrates that wake steering optimisation can exhibit model sensitivity due to the flatness of the Jensen model, while the Gaussian, GCH and Multizone models exhibit multi-modal objective functions. Both observations are of importance to understanding the behaviour and sensitivity of wake steering optimisation for the following cases of more complex farm layouts.

## 4.2 Multiple-turbine optimisation: $5 \times 5$ array of NREL 5 MW turbines

We consider wake steering optimisation for farm power maximisation on a $5 \times 5$ wind farm layout with NREL 5 MW turbines. Although the Jensen model was seen to perform poorly in Sect. 4.1, we still report its performance for this example due to the

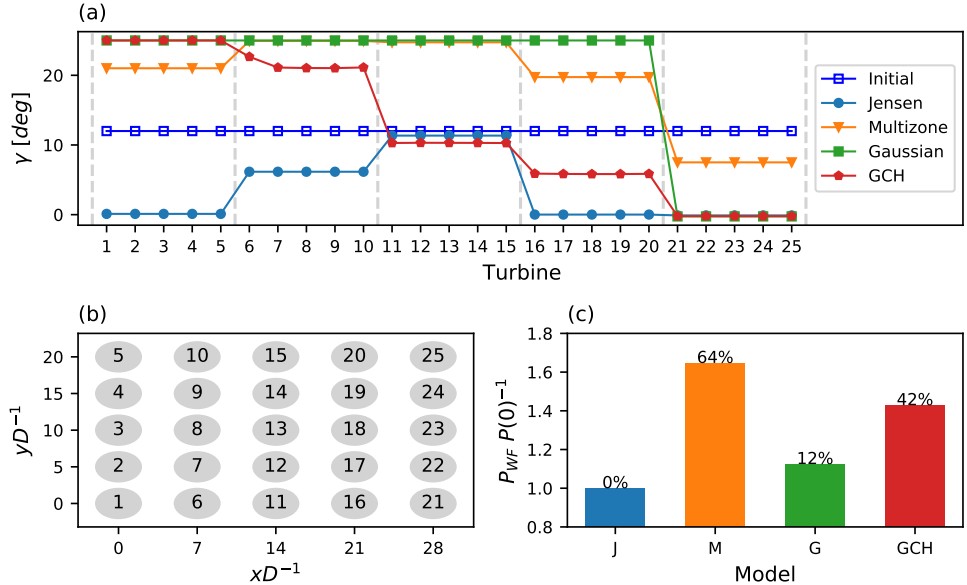

**Figure 3.** Model comparison for a $5 \times 5$ wind farm SLSQP optimisation. (**a**): optimal set-points per turbine, with $12\,°$ initial yaw angles illustrated in blue. Dashed lines delimit wind farm rows. (**b**): $5 \times 5$ wind farm layout with turbine naming convention. (**c**): resulting objective function (normalised farm power) for Jensen (J), Multizone (M), Gaussian (G) and Gauss-Curl Hybrid (GCH).

Jensen model's ongoing use in the wind energy community for wake steering applications (Kheirabadi and Nagamune, 2019; Houck, 2021; Andersson et al., 2021). The objective of this medium-complexity case is to understand the sensitivity of wake steering performance to both the choice of wake model, and to the choice and particular implementation of an optimisation algorithm. Due to the high-dimensional and multi-modal nature of the underlying optimisation problem in this case, statistical analysis is required to quantify optimisation sensitivity (see Sect. 4.2.3). However, we first study several indicative cases, which will render the statistical results transparent.

### 4.2.1 Initial yaw angle sensitivity in gradient-based optimisation

In this section, we seek to understand the interaction between the wake model selection, the algorithm choice, and the initial yaw angles used by each optimiser. Insights from the indicative examples given in this section will motivate and clarify the statistical analysis of Sect. 4.2.3.

Figure 3 shows the comparison between wake models for an SLSQP optimisation initialised with a yaw angle of $12\,°$ for all turbines. Significant discrepancies between models are observed for both predicted wind farm power improvement, and for the corresponding optimal yaw angles. Farm power variations of up to $64\,\%$ occur between models. Even the Gaussian and the GCH wake models have an optimised farm power difference of $30\,\%$ despite the similarity in these models' velocity deficit distribution (see Fig. 1).

Arguably more important, from an implementation perspective, is the significant inter-model variation in optimal yaw angles. For the Jensen model, Fig. 3 indicates non-zero optimal yaw angles for only the second (at 6 °) and third rows (at 9 °) of turbines. This counter-intuitive solution should be viewed in the context of a 0 % farm-power improvement, and the power-curve flatness described in Sect. 4.1. The Jensen model's lack of sensitivity implies that moderate yaw angle variations are essentially indistinguishable to the optimiser and, consequently, a gradient-based algorithm such as SLSQP can terminate at any point in a large set of possible yaw angles: those obtained in Fig. 3 are just one such near-optimal solution. Optimal yaw angle differences are also observed between the Gaussian model, which saturates the upper bound constraints at 25 ° for all but the final turbine row, and the GCH model, whose optimal angles are monotonically decreasing with turbine row. This is attributed to the GCH model's ability to capture secondary steering effects (King et al., 2021). Finally, the Multizone model's optimal yaw angles are non-monotonic with row, but also identify the non-physical setting of 8 ° for the final row of turbines (0 ° is optimal). As will be explored later in this section, this behaviour is caused by optimisation sensitivity to initial yaw angles.

To further explore the sensitivity to initial yaw angles between models, Fig. 4 shows the results of 10 SLSQP optimisation runs, each initialised by randomly sampling the 25 initial yaw angles from a uniform distribution on $[-25\,°, +25\,°]$, with these initial conditions then used for all four models. These optimisation runs will be incorporated into the statistical analysis detailed in Sect. 4.2.3. All models exhibit strong sensitivity to initial yaw angles. Interestingly, for Jensen and Gaussian models, while there is almost no variation in normalised farm power improvement, there is high variability in the optimal decision variables. As discussed in Sect. 4.1, for the Jensen model, this arises due to cost-function flatness, and for Gaussian, due to the symmetric multi-modality of its objective function. A further consequence of multi-modality, which can be seen in Fig. 4-c, is that yaw constraints may be saturated at either boundary (i.e. at $+25\,°$ or at $-25\,°$), rather than uniformly at $+25\,°$ as in the single case shown in Fig. 1. Further, this column-wise switching of optimal yaw angle signs has a negligible effect on the predicted farm power improvement. Such sign inconsistency is clearly undesirable for practical implementation of wake steering since small operational perturbations could potentially require large control input changes. This behaviour is discussed further in Sect. 4.2.3 and Sect. 4.3.

For the Multizone and GCH models, the sensitivity of both the normalised farm power (of up to 15 % and 12 %, respectively) and of the optimal yaw angles can be observed in Fig. 4. The Multizone model (Fig. 4-b) is highly sensitive to initialisation, caused by its highly non-linear and multi-modal farm power function—discussed in more detail in Sect. 4.2.2—which arises due to this model's definition of splitting the wake into various zones, with sharp interfaces between each zone (see Appendix A1.2 and Fig. 1-b). The GCH model (Fig. 4-d) has distinct local maxima, which is clear from the large differences observed in the normalised power increase across the 10 optimisation runs. Similar to the Gaussian model, multi-modality causes column-wise variation in the sign of the optimal decision variables in a number of runs (e.g., run "Random 10"). However, due to the fact that the GCH model also captures secondary-steering and wake rotation effects, such sign switches now also correspond to sub-optimal *local* maxima, with an observed range of 12 % in normalised farm power for even the small number of cases considered here.

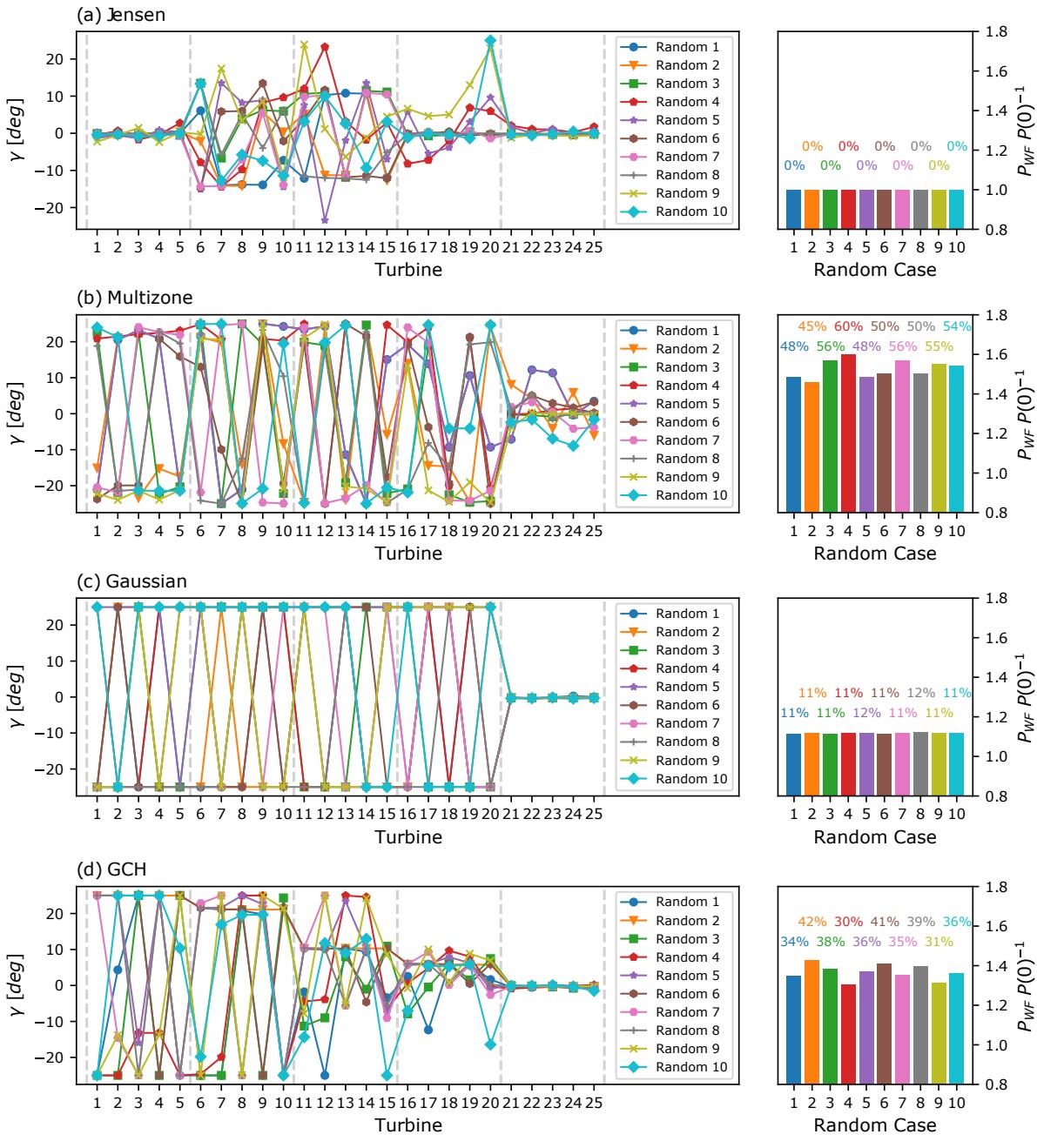

**Figure 4.** Initialisation sensitivity of a $5 \times 5$ wind farm SLSQP optimisation for the (**a**) Jensen, (**b**) Multizone, (**c**) Gaussian, and (**d**) GCH wake models. Left plot: optimal set-points per turbine for the random sets "Random" of initial conditions. Dashed lines delimit wind farm rows. Right plot: resulting objective function (normalised farm power) per initialisation case. For turbine naming convention, refer to Fig. 3-b.

In terms of wake model robustness, two key observations deserve attention. Firstly, the Gaussian model demonstrates significantly greater robustness than the Multizone model despite both models yielding similar optimal yaw settings (Fig. 4-b,c). The Gaussian model exhibits notably lower variations in farm power improvement across different initialisations. Secondly, although the GCH model (Fig. 4-d), which includes secondary steering effects, offers more physically accurate optimal yaw angles than the Gaussian model, it exhibits considerably lower robustness, as evidenced by significant variations in the resulting objective function based on the initial yaw angles.

To investigate initialisation sensitivity in more detail and to better understand how initial yaw angles affect the optimal yaw angles, we study one additional SLSQP optimisation run—denoted Test Case 1—for each of the considered models, again initialised by sampling 25 turbine yaw angles from a uniform distribution on $[-25\,°, +25\,°]$. The resulting optimal yaw angles are shown in Fig. 5-left, while the streamwise velocity at hub height for the optimal yaw configuration is shown in Fig. 5-right. It is clear that the initial yaw angles (blue squares) directly influence the optimal yaw angles (red circles) for the Jensen, Multizone and Gaussian models, both in their sign and absolute value. For the Gaussian model, for example, the sign of the initial angle for each turbine directly determines the sign of its optimised angle, which either saturates the constraints or is zero if the turbine is in the final row. This results in optimal yaw angles with within-column sign oscillations, which are practically undesirable. Their effect on the streamwise velocity fields is shown in Fig. 5-right.

For the GCH model in Fig. 5-d, the sign influence of initial yaw angles is still apparent, although less obvious than for the other models. As discussed above, the resulting within-column oscillations of optimal yaw angles correspond to local, not global, maxima. To quantify this observation, consider the five columns of turbines in Fig. 5-d-right for the GCH-based optimisation. The upper column, at $y \approx 2,500$ m, exhibits no sign fluctuations of optimal yaw angle within the same column. This is in contrast to the other four columns, which extract, respectively, $13\,\%$, $2\,\%$, $16\,\%$, and $7\,\%$ less power than the upper, sign-consistent, column. Understanding the origin of these local maxima, and why the gradient-based optimiser SLSQP identifies different maxima for different initial yaw values requires a more careful analysis of the underlying objective function geometries, conducted and presented in Sect. 4.2.2.

Figure 5-left also includes the resulting optimal yaw settings (green triangles) for the global optimisation algorithm TuRBO, introduced in Sect. 3.1. Compared to SLSQP, TuRBO appears to mitigate the influence of initial yaw angles on the sign and magnitude of the optimal yaw angles for all wake models. For the GCH model (Fig. 5-d), for example, the optimal yaw configuration exhibits mostly sign-consistent and positive yaw angles, with the exception of only turbines $1, 6$ and $11$. In further contrast to SLSQP, the optimal yaw angles obtained by TuRBO exhibit an approximate monotonically decreasing trend with downstream distance, which is representative of the optimal solution for models capturing secondary steering effects, as observed in King et al. (2021); Zong and Porté-Agel (2021).

These results motivate a statistical comparison of optimisation sensitivity between gradient-based and global optimisation approaches, which will be given in Sect. 4.2.3. Before this, we will briefly study the reasons for the increased sensitivity of gradient-based optimisation by looking more closely at the underlying geometry of the objective function.

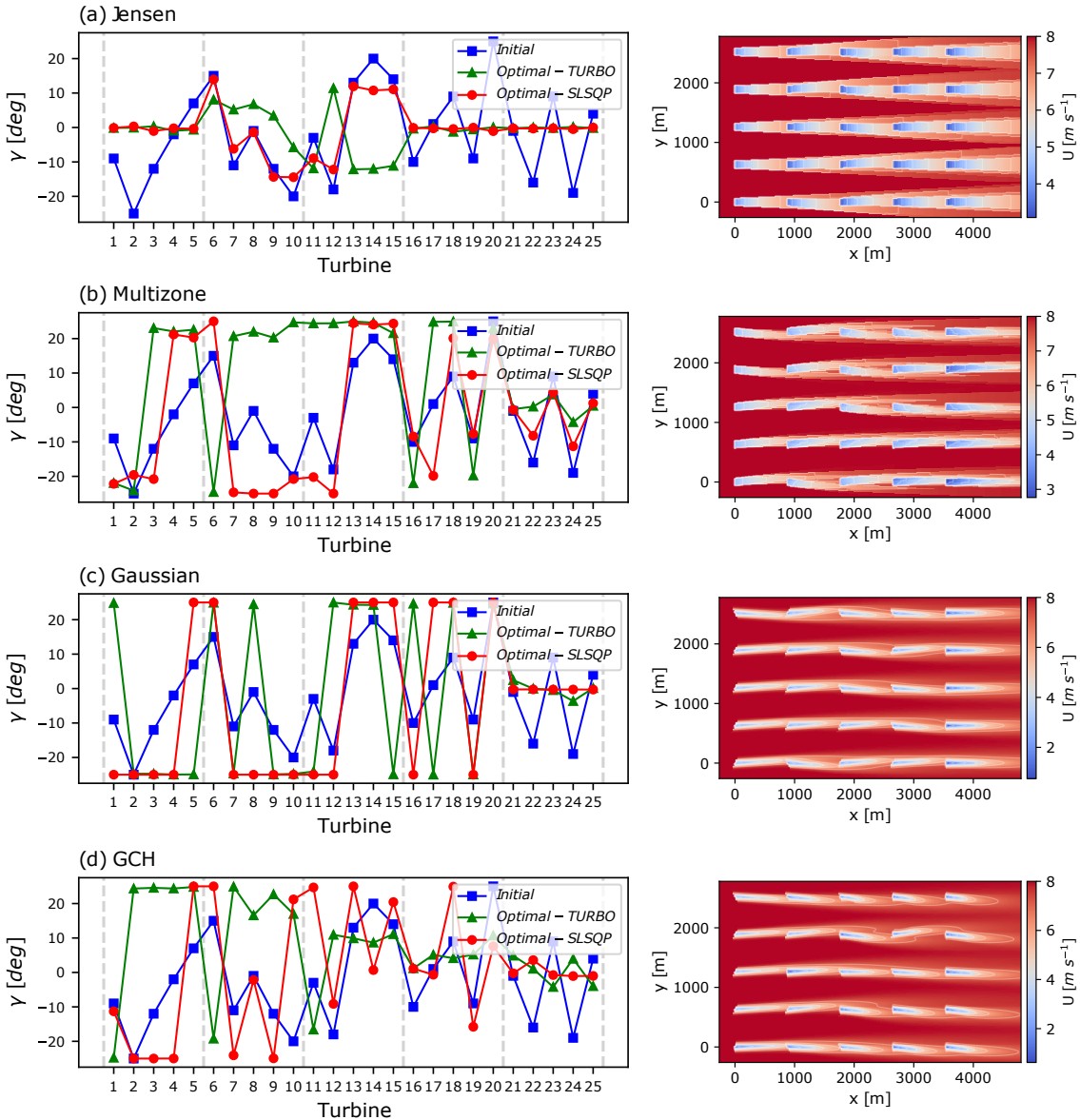

**Figure 5.** "Test Case 1" optimisation results of a $5 \times 5$ wind farm layout for the (**a**) Jensen, (**b**) Multizone, (**c**) Gaussian, and (**d**) GCH wake models. Left plot: SLSQP (in red circles) and TuRBO (in green triangles) optimised yaw angle per turbine given "Test Case 1" random set of initial points (in blue squares). Dashed lines delimit wind farm rows. Right plot: streamwise velocity at hub height for SLSQP optimised yaw set-points. For turbine naming convention, refer to Fig. 3-b.

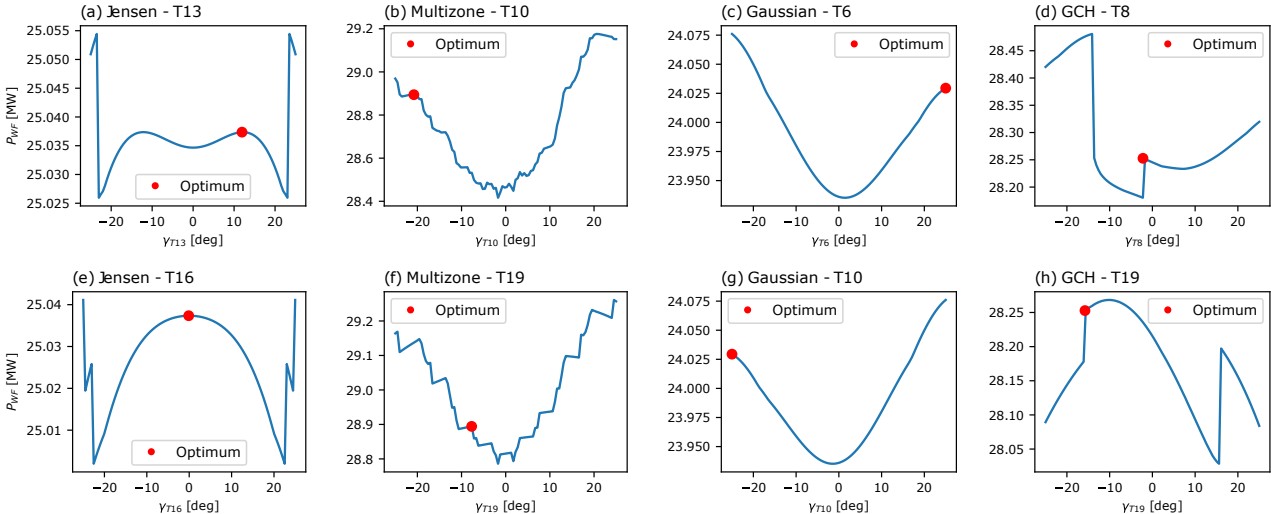

**Figure 6.** Farm power of a $5 \times 5$ wind farm layout for the (**a,e**) Jensen, (**b,f**) Multizone, (**c,g**) Gaussian, and (**d,h**) GCH wake models as a function of a single turbine yaw angle. Turbines yaw settings range $\pm 25\,°$, with SLSQP optimisation solutions as starting points (illustrated with red circles for the swept turbine "T"). For turbine naming convention, refer to Fig. 3-b.

### 4.2.2 Objective function geometry

This section investigates the geometry of the farm power objective function when using different wake models. The aim is to understand the higher initialisation sensitivity observed in gradient-based optimisation and provide the necessary insights to propose a solution to this problem, which is presented in Sect. 4.3.

Figure 6 shows, for each model, two planar slices through the objective functions $G(\boldsymbol{\gamma})$ for Test Case 1. Each is obtained by varying the yaw angle of a single chosen turbine (as indicated in each subfigure), with each presented slice also containing the optimal yaw angles obtained for each model. All four models exhibit local extrema, confirming the multi-modal nature of all farm power functions. Even the Gaussian model (Fig. 6-c,g), which has the smoothest objective function, has local maxima due to the optimisation constraints. The fact that the obtained optimum in this case (shown by red circles) is *not* global is now transparently explained by the fact that the initial yaw angles (shown in Fig. 5) of turbine T6 and T10 are of positive and negative sign, respectively.

The remaining three models have objective functions with either discontinuities or discontinuous derivatives. This is particularly prominent for the Jensen and Multizone models in Fig. 6-a,e and 6-b,f, respectively. This property is further exacerbated by the multiple wake interactions present for the $5 \times 5$ farm. As discussed in Sect. 3.1, a careful choice of optimisation parameter *eps* can help mitigate the effect of cost-function discontinuities. Since *eps* determines the step size used for finite-difference gradient approximations, using a larger value can numerically smooth the objective function. However, in this case, it can be seen that even a relatively large, and standard, choice of *eps* corresponding to a $2.5\,°$ step-size cannot overcome the observed

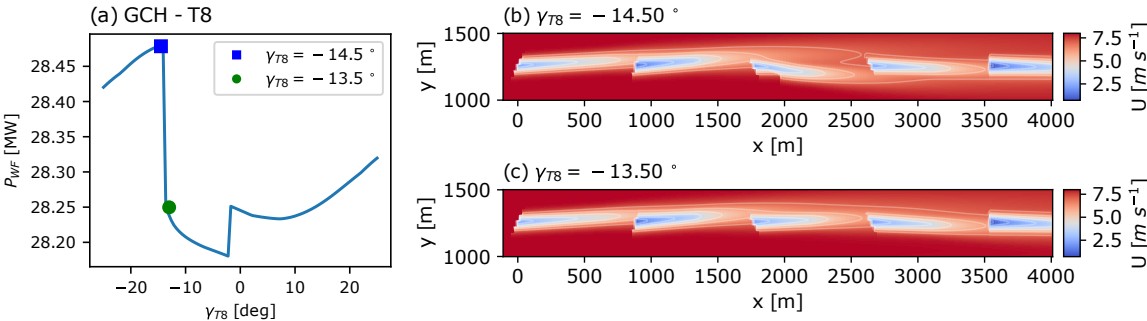

**Figure 7.** Farm power function discontinuity of a $5 \times 5$ wind farm layout for the GCH wake model. Left plot (**a**): wind farm power as a function of turbine "8" yaw angle, with the optimised solution as a starting point. Right plots: streamwise velocity at hub height of the third column of turbines for $\gamma_{T8} = -14.5\,^{\circ}$ (**b**) and $\gamma_{T8} = -13.5\,^{\circ}$ (**c**). For turbine naming convention, refer to Fig. 3-b.

cost-function roughness for Jensen, Multizone or GCH models (only local maxima are obtained). This suggests that an optimal value of *eps* may be highly problem dependent for non-smooth underlying models, and thus hard to identify a priori.

Finally, we further comment on the discontinuities present in the GCH model. These seem counter-intuitive given the GCH model's use of smooth velocity distributions, as shown in Fig. 1-d. However, the sharp discontinuities apparent in Fig. 6-d,h arise due to secondary-steering effects, where the wake of a downstream turbine in the GCH model can experience a sudden change in deflection due to the overlapping with the influence region of the vortices generated by one or more upstream turbines. Figure 7 confirms this behaviour by showing the flow field on either side of one jump discontinuity present in the cost-function slice of T8 in Fig. 6-d. Here, a small $1\,^{\circ}$ change in yaw angle of a single machine gives a $1\,\%$ loss in *total* farm power production.

The intrinsically multi-modal, and often rough, nature of farm power objective functions for the $5 \times 5$ farm presents a challenge for gradient-based optimisation. In Sect. 4.3 we provide a solution to this problem for gradient-based optimisers. However, we first consider the possible advantage of using a global optimisation strategy and perform a statistical comparison of optimisation sensitivity between gradient-based and global optimisation approaches for the $5 \times 5$ farm power maximisation problem.

### 4.2.3 Sensitivity comparison of gradient-based and global optimisation algorithms for wake steering

In this section, we compare the performance of the SLSQP and TuRBO optimisers described in Sect. 3.1 in terms of the statistics of two properties: the normalised farm power increase, and the total number of wind farm evaluations required. Statistical analysis is required since SLSQP is highly sensitive to initial conditions (see Sect. 4.2.1), and TuRBO is stochastic by nature. Moreover, computational complexity is compared in terms of farm evaluations rather than computational time because TuRBO requires the tuning of Gaussian processes. This additional computational step is highly dependent on the problem size, and including it in the comparison would lead to conclusions not generalisable to other optimisation cases. For

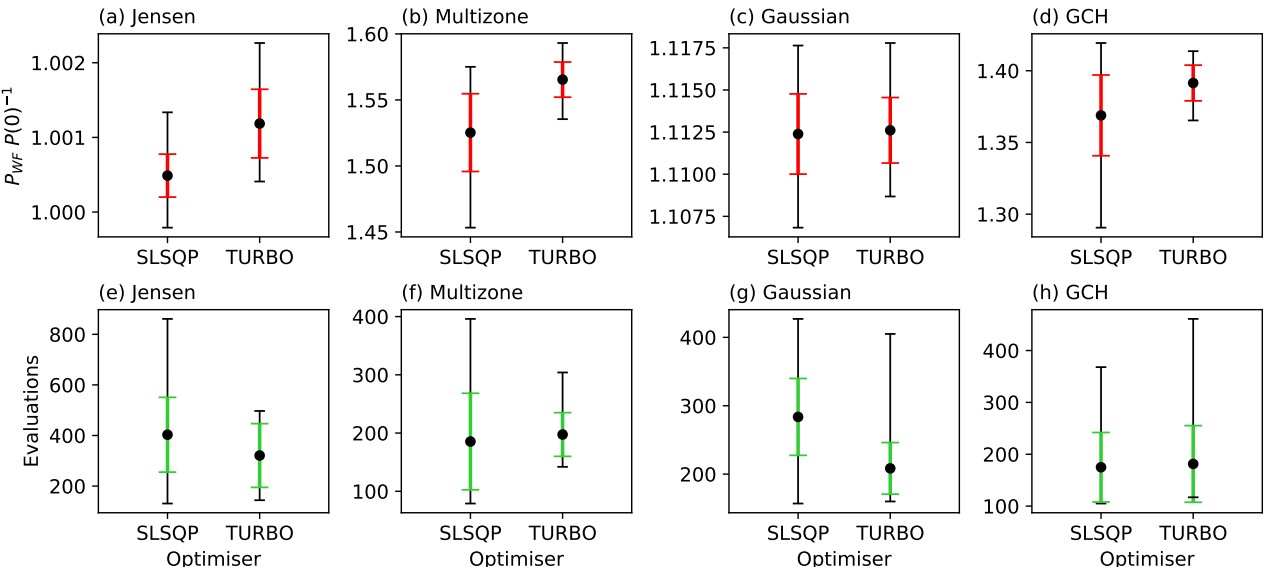

**Figure 8.** Comparison of the objective function (**a-d**) and the number of evaluations (**e-h**) statistics between SLSQP and TuRBO optimisation algorithms for a $5 \times 5$ wind farm layout using the (**a,e**) Jensen, (**b,f**) Multizone, (**c,g**) Gaussian, and (**d,h**) GCH wake models. Mean values are illustrated with error bars for one standard deviation (red or green) and minimum and maximum values (black).

each of the four models, wake steering optimisation is repeated 50 times per optimiser, each initialised with an independent set of random initial conditions sampled from a uniform distribution on $[-25\,°, +25\,°]$. To ensure a fair comparison, SLSQP and TuRBO algorithms share the same sets of initial yaw angles. The maximum number of evaluations for TuRBO is set to 500, corresponding to the observed average number of objective function evaluations used by SLSQP. Finally, to limit the bias of extreme cases in the function evaluation statistics, each algorithm is assumed to be converged when 95 % of the actual wind farm power improvement is reached.

Figure 8's upper row shows the comparison between SLSQP and TuRBO with the mean, standard deviation, minimum and maximum values of normalised farm power increase shown. Higher mean values are obtained for all models by the global TuRBO algorithm. Although the balance between exploration and exploitation can represent a challenge when using global optimisers, especially in highly dimensional search spaces, the multi-modal nature of the farm power function clearly gives an advantage to TuRBO's global approach for this $5 \times 5$ farm configuration. Moreover, for all models except Jensen, a lower standard deviation in the obtained optimal farm power is found when using TURBO, indicating that it is less sensitive to yaw angle initialisation. Realistic wind farm operating conditions also include slightly misaligned cases. For this reason, we also considered small $\pm 2°$ variations in the wind direction. The results, which are not included for the sake of brevity, were found to be consistent with the ones in Fig. 8.

Statistics (mean, standard deviation, minimum and maximum values) of the number of evaluations required for each optimiser are shown in the lower row of Fig. 8. For all the wake models, the TuRBO algorithm requires roughly the same or lower

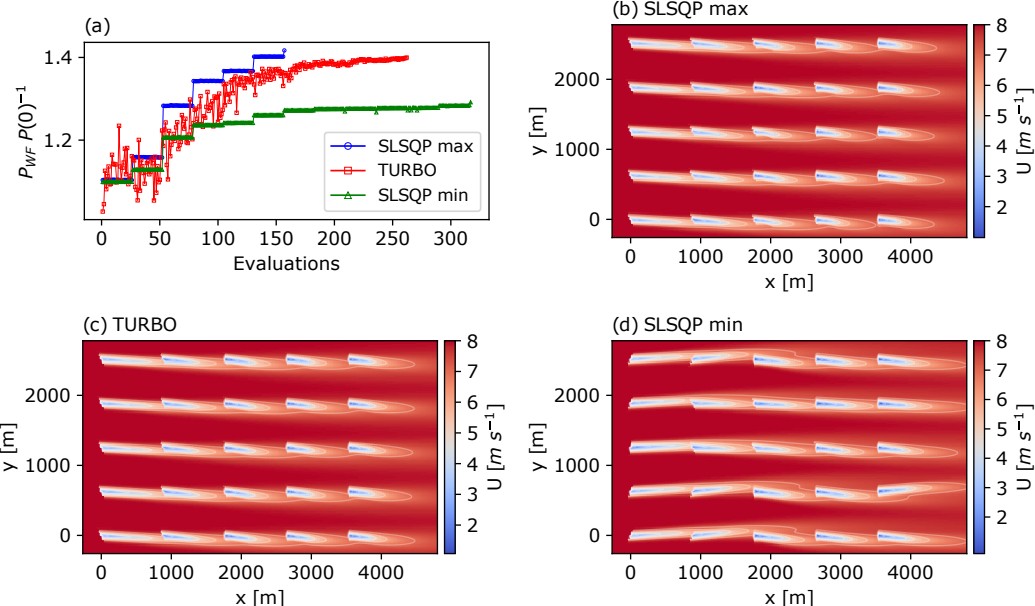

**Figure 9.** Comparison of optimisation results of a $5 \times 5$ wind farm layout between the SLSQP and TuRBO optimisation algorithms for the GCH wake model. Plot (**a**): objective function values at each evaluation for SLSQP maximum (blue circles), average TuRBO (red squares), and SLSQP minimum (green triangles). Remaining plots: resulting streamwise velocity at hub height with optimised yaw set-points for SLSQP maximum (**b**), average TuRBO (**c**), and SLSQP minimum (**d**) farm improvements.

evaluations than SLSQP to achieve, on average, higher wind farm power improvements. Moreover, TuRBO exhibits lower or similar standard deviations than SLSQP, indicating a lower impact of initialisation on computational requirements.

Figure 9 presents three indicative cases to understand the comparison of SLSQP and TuRBO optimisers for the GCH model. These cases include the TuRBO run achieving a farm power production closest to TuRBO's mean power increase, and the two SLSQP runs corresponding to the minimum and the maximum SLSQP farm power increases, respectively. Figure 9-a shows the objective function $G(\boldsymbol{\gamma})$ whenever a farm evaluation is performed by the respective optimiser. It is interesting that the best-performing SLSQP run (blue circles) converges after fewer iterations, and to a higher farm power, than the indicative

TuRBO case presented and, in fact, outperforms all TuRBO runs (see Fig. 8-d). This suggests that, if initialised in the region of attraction of a global (or near-global) maxima, a gradient-based optimiser may exhibit faster convergence than a generic global strategy.

The streamwise velocity flow fields at hub height for the three considered cases are also shown in Fig. 9. Inter-column switches of optimal yaw angles are again present in the worst performing SLSQP case (see Fig. 9-d, and green triangles in

Fig. 9-a), which requires $88\,\%$ more objective function evaluations and has a $10\,\%$ drop in farm power compared to the best-performing SLSQP case (see Fig. 9-b, and blue circles in Fig. 9-a). This again highlights the challenge of multimodality for gradient-based approaches. Finally, it is interesting to note that while the optimal wake velocity fields in Fig. 9-b and 9-c are

visually similar, they are not identical: the TuRBO solution has 1.5 % lower farm power, and requires 55 % more evaluations to find it. This confirms a possible, albeit counter-intuitive, advantage of the rapid local convergence enjoyed by gradient-based optimisers. Indeed, the quasi-Newton algorithm employed in SLSQP is well-known (Deuflhard, 2011) to possess rapid locally-quadratic convergence rates, which may partially explain this observation. It should be noted, however, that the high variability of SLSQP seen in Fig. 8, both in terms of farm power increase and objective function evaluations, is a barrier to obtaining robust performance advantages of SLSQP over global optimisation approaches such as TuRBO.

Although initialisation sensitivity is reduced using TuRBO's global optimisation strategy, within-column sign inconsistency of the optimal yaw angles is still possible for all wake models using both SLSQP and TuRBO algorithms, as shown in Fig. 5. This behaviour is practically undesirable since it suggests that large control input changes may arise from small parametric changes (e.g., if a small change in atmospheric conditions requires recalculation of the optimal yaw angles, a different distribution of within-column signs could be obtained, which would require a large change in yaw angles). As mentioned in Houck (2021), an excessive yawing is undesirable and should be limited whenever possible. For wake models describing power asymmetry due to yaw (Multizone and GCH), inter-column switches of optimal yaw angles also correspond to local maxima, with losses in farm power improvement. In the following section, we show how this issue can be overcome by adding simple constraints which reduce the initial yaw angle sensitivity of the SLSQP algorithm to enable rapid and robust convergence to desirable local maxima.

### 4.3   Multiple-turbine case: the Horns Rev Wind Farm

In this section, the results for farm power maximisation via wake steering optimisation are presented for the well-known Horns Rev wind farm. To the best of our knowledge, only a small number of studies, see for instance Dou et al. (2020); Zong and Porté-Agel (2021); Chen et al. (2022), have performed wake steering optimisation on this farm layout. In Zong and Porté-Agel (2021), sign coherent optimal yaw settings are obtained using gradient-based optimisation. However, in this study, explicit knowledge of the optimal solution was used to define initial yaw angles to ensure optimiser convergence. In contrast, Dou et al. (2020) used an evolutionary optimisation algorithm and found optimal yaw angles whose signs exhibit high column-inconsistency, reflecting the observations made for the $5 \times 5$ layout considered in Sect. 4.2.3.

The first objective of this high-complexity optimisation case is to confirm whether wake steering sensitivities uncovered in the low and medium complexity cases transfer to a realistic setup. The second objective is to add optimisation constraints to try to mitigate initial yaw angle sensitivity and to achieve coherent and interpretable optimal yaw settings that could eventually be implementable in the field. In large-scale optimisation problems with many design variables and required evaluations, the TuRBO algorithm becomes computationally demanding for real-time control applications due to the tuning of the Gaussian processes. For the Horn Rev case, TuRBO's computational complexity is about two orders of magnitude higher than SLSQP. For this reason, and the observation from Sect. 4.2.3 that SLSQP can exhibit rapid convergence if the initialisation sensitivity is appropriately mitigated, we only employ the SLSQP optimisation algorithm in this section, using the parameters in Table 2. Note that the Jensen model is not included in this section due to the poor wake steering performance observed in the previous sections.

For sensitivity mitigation, we consider two additional sets of optimisation constraints. The constraint "C1" refers to the case in which the yaw angles of every turbine are constrained to be positive. The aim is to improve optimiser performance by avoiding the maxima corresponding to within-column yaw angles with alternating signs. In aligned conditions, constraining the yaw angles to be either positive or negative is equivalent for symmetric models like the Jensen and the Gaussian. However, when considering wake models that incorporate wake rotation effects, such as the Multizone and the GCH models, positive yaw angles lead to higher turbine power production. Implementation of the constraint "C1" involves modifying the variable bounds from $[-25\,°, +25\,°]$ to $[0\,°, +25\,°]$. To maintain a step-size of $2.5\,°$ for gradient approximations, the SLSQP parameter $eps$ is adjusted to 0.1, as outlined in Table 2. A second set of constraints, "C2", corresponds to the case where the yaw angles of each turbine (excluding turbines in the most upstream row) are forced to be equal to or lower than the yaw angles of the upstream turbines in the same column, and is implemented in SLSQP using linear inequality constraints. It should be noted that these additional sets of constraints can be easily adapted to other operating and atmospheric conditions, e.g. for a different wind direction, a simple permutation of the turbines' labelling is required to specify columns which correspond to the new downstream direction.

The investigated cases are limited to a fully aligned layout, as it is the predominant condition and the one holding the largest potential for the implementation of wake steering strategies. The analysis performed is statistical, where each of the 50 optimisation cases per wake model is initialised with an independent set of random initial conditions sampled from a uniform distribution on $[-25\,°, +25\,°]$. For clarity throughout the section, optimisation performed using only the original yaw angle bounds of $[-25\,°, +25\,°]$, without applying either constraint "C1" or "C2", is referred to as "nominal". Optimisation cases with additional constraints are indicated as "constrained". Note that for the additional constrained Horns Rev cases, a $2.5\,°$ step-size corresponds to $eps = 0.1$ due to the modified variable bounds $[0\,°, +25\,°]$, as specified in Table 2.

Optimal yaw configurations for a single SLSQP optimisation with a unique random set of initial conditions in the nominal, "C1", and "C1+C2" optimisation cases are presented in Fig. 10. In line with the results of Dou et al. (2020), the first row of subfigures shows that, in the nominal case, the optimal yaw settings obtained for the three wake models are not practical, with frequent switches between large positive and negative yaw angles within any given row or column of the wind array. The only common trend is that the final downstream row of turbines has optimal yaw angles close to zero. The key message is that, without a sensitivity mitigation strategy, the proposed yaw angle settings cannot be implemented in real life due to obvious operational constraints.

The introduction of additional constraints clearly improves the identification of the optimal variables. For all models, optimal yaw angles represent a more interpretable and practical solution. Overall, similar performance is achieved with the two different constrained approaches, "C1" and "C1+C2". This successfully proves how the use of simple constraints, motivated by an understanding of the optimisation problem, can lead to considerable improvements in optimisation performance. The "C1" constraint forces the optimiser to avoid the maxima corresponding to within-column yaw angles with alternating signs, of particular importance when such maxima are sub-optimal for models with power asymmetry due to yaw (Multizone and GCH). The "C2" constraint, limited to "C1+C2" cases, effectively decreases initialisation sensitivity by imposing column equal or monotonic decreasing optimal yaw angle solutions. For the GCH model, which captures secondary steering effects,

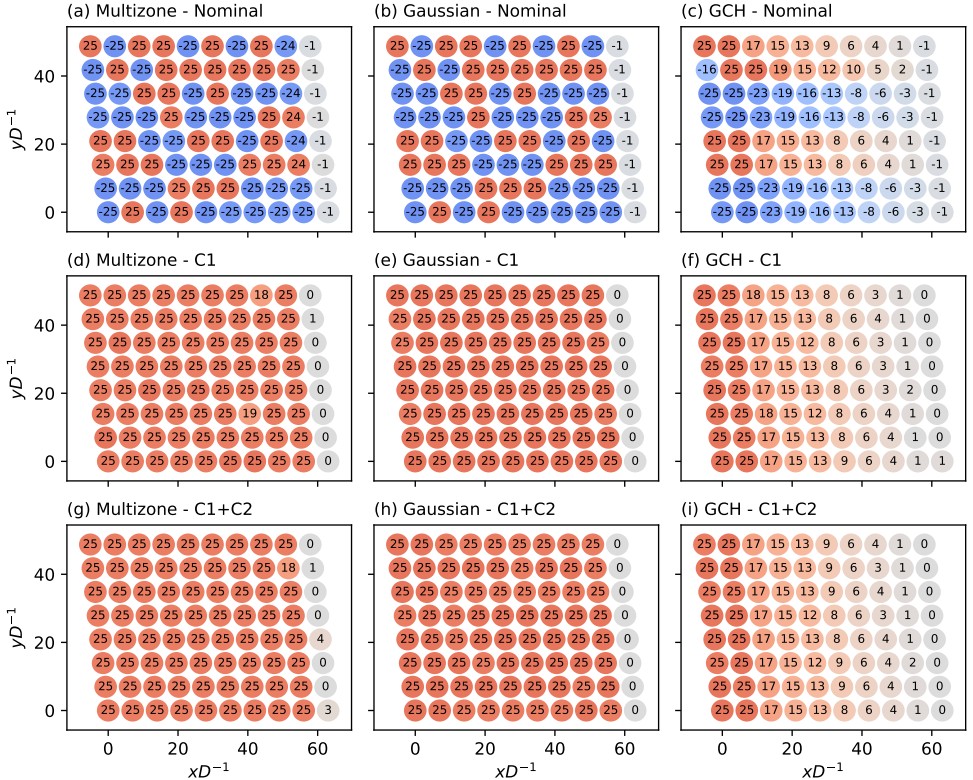

**Figure 10.** Comparison of the Horns Rev wind farm optimal yaw setting for a single optimisation between nominal, "C1", and "C1+C2" constrained SLSQP optimisation cases for the (**a,d,g**) Multizone, (**b,e,h**) Gaussian, and (**c,f,i**) GCH wake models. Red and blue indicate positive and negative yaw angles, respectively.

introducing constraints leads to a smoother decrease in optimal decision variables from row to row, and to the identification of an optimal decrease rate.

Table 3 provides the farm power improvements obtained from the single SLSQP optimisation outlined in Fig. 10 for the nominal, "C1", and "C1+C2" optimisation cases. Across all three wake models, it is evident that the introduction of additional

**Table 3.** Comparison of the farm power improvements for a single Horns Rev wind farm optimisation between nominal, "C1", and "C1+C2" constrained SLSQP optimisation cases for the Multizone, Gaussian, and GCH wake models.

| Case name | Multizone $P_{WF}P_{WF}(0)^{-1}$ | Gaussian $P_{WF}P_{WF}(0)^{-1}$ | GCH $P_{WF}P_{WF}(0)^{-1}$ |
|---|---|---|---|
| Nominal | 1.71 | 1.11 | 1.57 |
| "C1" | 1.82 | 1.12 | 1.63 |
| "C1+C2" | 1.83 | 1.12 | 1.63 |

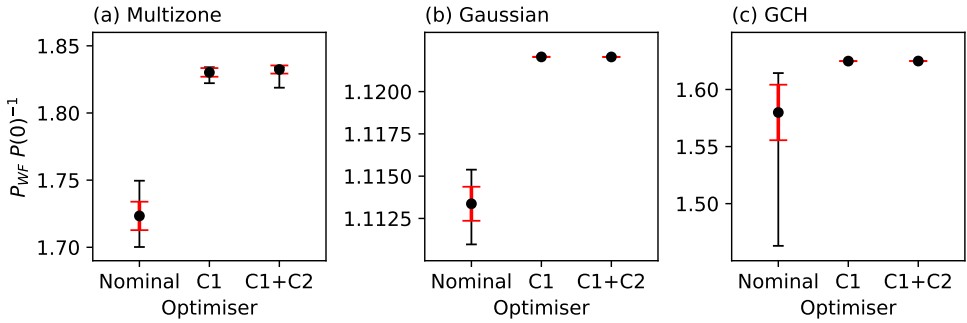

**Figure 11.** Comparison of the Horns Rev wind farm objective function statistics between nominal, "C1", and "C1+C2" constrained SLSQP optimisation cases for the (**a**) Multizone, (**b**) Gaussian, and (**c**) GCH wake models. Mean values are illustrated with error bars for one standard deviation (red) and minimum and maximum values (black).

constraints enhances the performance of the optimiser. Compared to the nominal case, the constrained cases yield a $7\%$, $1\%$, and $4\%$ increase in the objective function for the Multizone, Gaussian, and GCH models, respectively. When comparing the "C1" and "C1+C2" optimisation cases, minimal or negligible differences in farm power improvement are observed. This indicates that the introduction of the "C1" constraint alone already leads to column equal or monotonically decreasing optimal

yaw angles in the cases presented (as depicted in Fig. 10). However, to consistently ensure this behaviour across various optimisation conditions, the implementation of the "C2" constraint is necessary. While examining the results of a single SLSQP optimisation is informative, a statistical analysis is required to evaluate the benefits of implementing additional constraints to mitigate initialisation sensitivity.

     Figure 11 presents the statistical results for farm power improvement in the nominal, "C1", and "C1+C2" optimisation cases.

For all wake models, a significant reduction in initialisation sensitivity can be clearly observed when constraints are imposed, both in terms of standard deviation and minimum and maximum values. For the GCH model, dependency on initial conditions in the nominal case leads to sub-optimal solutions with potential farm power losses up to $15\ \%$ (the difference between the maximum and minimum power increase values). With the introduction of constraints, these losses are below $0.1\ \%$. Further, a higher mean farm power improvement is experienced in the constrained cases for all models. In the Multizone case, avoiding

sub-optimal solutions due to constraints leads to a $12\ \%$ increase in mean farm power improvement. Finally, consistent with the conclusions for the medium-complexity $5 \times 5$ case, an overall strong dependency of wake steering strategies to wake model choice is observed, exhibiting significant discrepancies in farm power improvement. To further explore model sensitivity, an inter-model cross-validation is performed with the Multizone and Gaussian models' performance evaluated with the optimal decision variables computed using the GCH model. As expected, both models exhibit lower normalised farm power increases

when using the optimal GCH yaw angles ($24\%$ and $6\%$ lower for the Multizone and Gaussian models, respectively). These results suggest that the row-monotonic decrease in optimal yaw angles, attributed to the GCH model capturing secondary steering effects (King et al., 2021), represents a better solution to the wake optimisation problem than is possible when using

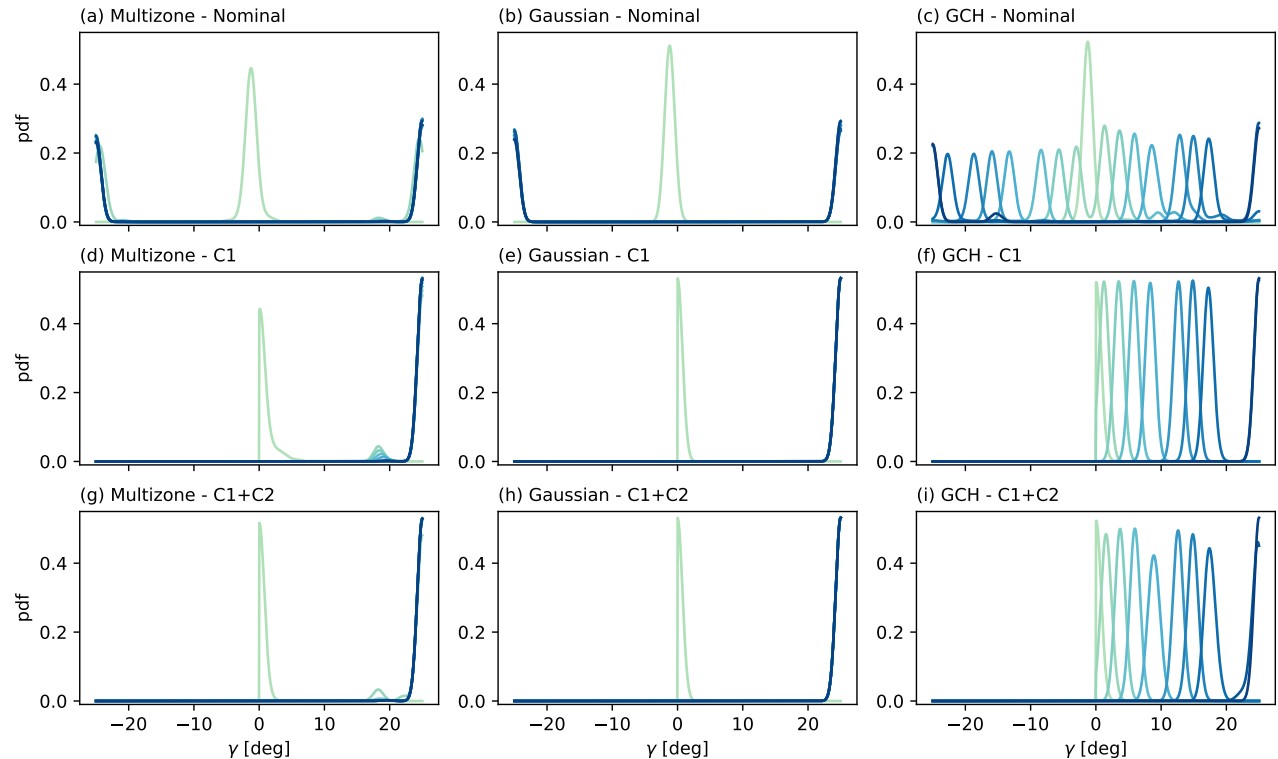

**Figure 12.** Comparison of the Horns Rev wind farm optimal yaw setting statistics between nominal, "C1", and "C1+C2" constrained SLSQP optimisation cases for (**a,d,g**) Multizone, (**b,e,h**) Gaussian, and (**c,f,i**) GCH wake models. Row-averaged probability density functions of the optimal yaw angles are shown, with a transition from dark blue (most upstream row) to light green (most downstream row).

either the Gaussian or Multizone models. This observation is in agreement with several recent studies, including King et al. (2021); Zong and Porté-Agel (2021); Howland and Dabiri (2021); King et al. (2022).

Figure 12 presents the statistical results for the optimal yaw settings in the nominal, "C1", and "C1+C2" optimisation cases. For each row in the farm layout, a probability density function of the optimal yaw angles is shown in this figure. It is defined by averaging the probability density functions for the single turbines in the row. The aim is to express the distributions of optimal yaw settings per farm row and see how they compare for different optimisation cases. When examining Fig. 12-d,e,f and Fig. 12-g,h,i for the "C1" and "C1+C2" constrained cases, respectively, it becomes evident that the probability density functions 530     exhibit higher magnitudes compared to the nominal cases shown in Fig. 12-d,e,f. With the exception of the last turbine row, the probability density function values, which are approximately 0.25 in the nominal cases, are at least twice higher when the additional constraints are introduced in the optimisation. This behaviour is observed across all wake models and indicates a more consistent distribution of optimal yaw angles for different initialisations, thereby highlighting improved optimisation robustness and reduced dependence on initial conditions for the constrained cases. For the Multizone and Gaussian models, 535     optimal yaw settings in the nominal case tend to converge at the optimisation bounds, as shown in Fig. 5 and confirmed by

bi-modal probability density functions in Fig. 12-a and 12-b. Caused by initialisation sensitivity and lack of constraints, this undesirable behaviour of bi-modality is easily removed in the "C1" and "C1+C2" optimisation cases. For the GCH model, bi-modal probability density functions can also be identified in the nominal optimisation results. However, the peaks of the modes exhibit a more complex pattern depending on the farm row (Fig. 12-c). As for the previous models, bi-modality is removed by the introduction of constraints. Finally, consistently with the discussion for the $5 \times 5$ farm layout in Sect. 4.2, significant overall discrepancies between wake models can be identified in terms of optimal yaw angles.

The presented results indicate that the simple strategy proposed for SLSQP sensitivity mitigation is effective for this realistic and complex wake steering optimisation problem. The enforcement of constraints successfully decreases the impact of initial conditions both in terms of the objective function and optimal decision variables, while achieving generally higher farm power improvements and more consistent optimal yaw settings. We note finally that the constrained gradient-based approach can be simply adapted to misaligned wind directions. This can be achieved by permuting the turbine labelling to create columns aligned with the incoming wind direction. Either constraint "C1" or "C1 + C2" can then be applied analogously in this situation.

## 5   Conclusions

A detailed investigation of the sensitivity of wake steering optimisation for increased power output of wind farms was carried out in this study. Sensitivity to the choice of analytical wake models, optimisation algorithm, and different operating conditions was assessed in terms of variability of both optimised farm power and optimal yaw angles. The study was performed with four different analytical wake models, with a gradient-based and a global optimisation algorithm. Three wind farm layouts were investigated: two generic layouts with $2 \times 1$ and $5 \times 5$ turbines, and a layout corresponding to the Horns Rev wind farm with 80 turbines. This hierarchy of test-cases with increasing complexity facilitated the understanding of the different sensitivities present in wake steering optimisation. By conducting analysis on initialisation sensitivity, the nature of the farm power function, and optimiser statistical performance, this investigation provided insight into the impact of modelling, algorithmic, and parametric choices on wake steering strategies to increase the power output of wind farms.

For the $2 \times 1$ layout, optimisation was performed by a simple parametric sweep, removing any sensitivity to algorithmic choice or implementation. The results indicated a strong sensitivity of wake steering strategies to the choice of analytical wake model and highlighted the role of flatness and multi-modality of the objective function in wake steering optimisation performance.

In the medium-complexity $5 \times 5$ case, optimisation was first performed using the gradient-based SLSQP optimisation algorithm. High optimisation sensitivity to the wake model choice, both in terms of objective function and optimal decision variables, was observed. All models exhibited strong variability of the objective function and optimal yaw angles to initial conditions, with sub-optimal solutions leading to potential power improvement losses. A comparison between the gradient-based SLSQP and the global optimiser TuRBO showed that, overall, TuRBO required fewer wind farm evaluations and achieved larger improvements in wind farm power production, with a reduced sensitivity to initial conditions. Although the multi-modal and discontinuous nature of the farm power function strongly affected the SLSQP performance, results in Fig. 9 also high-

lighted that rapid convergence could be achieved if initialisation sensitivity was reduced, motivating the sensitivity mitigation strategy proposed in the final test-case considered.

Finally, a higher complexity optimisation was performed with the SLSQP algorithm for the farm layout corresponding to the Horns Rev wind farm (80 turbines). Optimal yaw angles exhibited highly column-inconsistent signs, making them unsuitable to implement in practice. It was found that the introduction of constraints during the optimisation dramatically decreased the sensitivity to initial conditions while achieving more consistent and interpretable yaw settings. This work, therefore, demonstrates that the sensitivity of wake steering optimisation can be mitigated either by the use of global optimisation algorithms or by the addition of simple constraints to gradient-based algorithms. The success of a constrained gradient-based approach is especially promising, particularly with a view towards adaptation to new problems, due to the simplicity and transparency of gradient-based algorithms in contrast to more complex and less interpretable global strategies.

Future work will look at sensitivity of wind farm wake steering optimisation for a range of wind directions and inflow speeds, and investigate the impact of atmospheric conditions for more complex optimisation problems, in addition to identifying appropriate optimisation constraints to enable sensitivity mitigation in each setting. Future research will also focus on determining wake steering sensitivities in an online control framework, including state estimation and parameter tuning algorithms.

*Acknowledgements.* The authors would like to acknowledge the Department of Aeronautics, Imperial College London, for supporting this work funded by a doctoral studentship.

**Appendix A**

**A1   Wake models**

The most relevant parameter in analytical wake models is the streamwise velocity $u$ ($u_w$ in Eq. (1)) in the downstream region of a turbine subject to a mean free stream velocity $U_\infty$. Due to turbine operation and the consequent wake streamwise velocity deficit $\delta u$, the streamwise velocity is defined as

$$u = U_\infty(1 - \delta u) . \tag{A1}$$

Considering the unsteadiness of the wake phenomenon and its evolution in three-dimensional space, in principle, all terms in Eq. (A1) are a function of space and time $(x, y, z, t)$. However, the main assumption of analytical models is the description of a steady-state, where time dependency is ignored, and the focus is on equilibrium conditions. The model formulations in this section are presented based on velocity deficit $\delta u(x, y, z)$.

**A1.1   Jensen Model**

The Jensen model (Jensen, 1983) is a simple steady wake model widely used in the industry. It is derived by applying mass conservation to a control volume in the wake region of the wind turbine and relating the velocity deficit just behind the rotor to

the thrust coefficient $C_T$ using Betz theory. Hence, turbines are assumed to be actuator disks with rotor diameter $D$ and thrust coefficient $C_T = 4a(1-a)$, where $a$ is the turbine induction factor. The Jensen model is based on two main assumptions: the conservation of the cross-stream integral of the streamwise velocity deficit as the wake linearly expands downstream and the velocity deficit being simply a function of the downstream distance $x$. The first main assumption implies a quadratic decay of the velocity deficit with the linear expansion of the wake. The second one implies a uniform velocity deficit in the wake, hence an axisymmetric wake with a well-defined edge (see Fig. 1-a).

The streamwise velocity deficit induced by a turbine with diameter $D$, assumed to be operating at an induction factor $a$, can be expressed as

$$\delta u(x,r) = \begin{cases} 2a\left(\frac{D}{D+2kx}\right)^2, & \text{if } r \leq \frac{D+2kx}{2} \\ 0, & \text{otherwise} \end{cases}, \tag{A2}$$

where a cylindrical coordinate system is defined with the origin at the rotor hub of the first upstream turbine. Radial and streamwise distances are expressed by $r$ and $x$, respectively, and $k$ represents the dimensionless expansion coefficient.

The Jensen model is based on a steady description of the wake where turbines are modelled as actuator disks with uniform loading, and no notion of added turbulence intensity due to upstream turbines' operation is included. Moreover, it does not conserve momentum, it is limited to far wake predictions, and no calculation involving cross-stream and vertical velocity components is included. Despite these limitations, the Jensen model is simple and inexpensive. It can be used for control and optimisation studies, and it can provide valuable insights into power production for large wind farm layouts in normal operating conditions.

### A1.2  Multizone Model

The Multizone model, developed by Gebraad et al. (2014), is a modification of the Jensen model to better describe wake velocity profiles and partial wake overlapping, especially in yawed conditions. Within a turbine wake, three zones $q$ are defined: near wake zone ($q = 1$), far wake zone ($q = 2$), and mixing wake zone ($q = 3$). They are assumed to expand linearly with downstream distance $x$. Different expansion rates are determined based on the tuned model parameters $k_e$ and $m_{e,q}$. The wake diameter $D_{w,q}$ for each zone is defined as

$$D_{w,q}(x) = \max\left(D + 2k_e m_{e,q} x, 0\right) . \tag{A3}$$

The velocity deficit in the three wake zones is assumed to decay quadratically with the downstream distance rather than being directly related to the wake expansion as in the Jensen model. Due to the presence of different wake zones, a smoother transition from the wake centre to freestream velocity is achieved (see Fig. 1-b), where the velocity deficit spatial variables are the radial position from the wake centre $r$ and turbine downstream distance $x$. The wake remains axisymmetric. Considering a turbine assumed to be operating at an induction factor $a$ and yaw angle $\gamma$, the mean streamwise velocity deficit $\delta u(x,r)$ can be defined as

$$\delta u(x,r) = 2ac(x,r) , \tag{A4}$$

where $c(x, r)$ is the wake decay coefficient

$$c(x, r) = \begin{cases} c_1 & \text{if} \quad r \leq D_{w,1}/2 \\ c_2 & \text{if} \quad D_{w,1}/2 < r \leq D_{w,2}/2 \\ c_3 & \text{if} \quad D_{w,2}/2 < r \leq D_{w,3}/2 \\ 0 & \text{if} \quad r > D_{w,3}/2 \end{cases} . \tag{A5}$$

The local wake decay coefficient $c(x)$ for each wake zone $q$ is expressed by

$$c_q(x) = \left[ \frac{D}{D + 2k_e m_{U,q}(\gamma) x} \right]^2 , \tag{A6}$$

where $k_e$ parameter determines wake zones expansion and velocity deficit recovery, while $m_{U,q}$ is an empirically derived coefficient. For each wake zone, $q$, $m_{U,q}$ is defined as

$$m_{U,q}(\gamma) = \frac{M_{U,q}}{\cos(a_U + b_U \gamma)} , \tag{A7}$$

with the empirical parameters $M_{U,q}$ (ensuring outer zones faster recovery), $a_U$, and $b_U$ tuned with high-fidelity simulations.

The Multizone wake model is a computationally inexpensive model, suitable for control and optimisation studies, including yaw applications. However, it only describes equilibrium conditions, and it is limited to far wake predictions. In addition, it does not exhibit any sensitivity to inflow turbulence intensity, and it does not include considerations for added turbulence intensity by upstream turbines. Moreover, it involves many tuned empirical parameters, decreasing the model confidence for a various range of operating conditions. Finally, it does not explicitly conserve momentum.

### A1.3 Gaussian Model

The Gaussian wake model was originally developed by Bastankhah and Porté-Agel (2014) and recently improved by various studies in the literature (Abkar and Porté-Agel, 2015; Bastankhah and Porté-Agel, 2016; Niayifar and Porté-Agel, 2016; Dilip and Porté-Agel, 2017). This steady wake model consists of a mass and momentum conserving formulation based on a simplification of the Navier-Stokes equations. A Gaussian distribution on the streamwise velocity deficit is imposed by applying the self-similarity theory used in shear flows (see Fig. 1-c). Moreover, a linear wake expansion is assumed. The model accounts for atmospheric stability and added turbulence intensity due to upstream turbines. Defining a three-dimensional coordinate system at the rotor hub of the turbine, the streamwise velocity deficit in the far wake, dependant on the three spatial dimensions $(x, y, z)$, is defined as

$$\delta u(x, y, z) = C e^{-(y-\delta)^2/2\sigma_y^2} e^{-(z-z_{\text{h}})^2/2\sigma_z^2} , \tag{A8}$$

where $C$ is the streamwise velocity deficit at the wake centre, $\delta$ the wake deflection (refer to Appendix-A2), $z_h$ the hub height of the turbine, and $\sigma_y$ and $\sigma_z$ the standard deviations of the Gaussian velocity deficit at each streamwise location, indicating the width of the wake in the cross-stream and vertical direction, respectively. The general concept is to apply a Gaussian

distribution to the streamwise velocity deficit at the wake centre, given linearly increasing cross-stream and vertical wake widths and a cross-stream deflection due to turbine yaw misalignment. The wake centre velocity deficit $C$ is defined as

$$C(x) = 1 - \sqrt{1 - \frac{C_T \cos\gamma}{8\left(\sigma_y \sigma_z / D^2\right)}} \ , \tag{A9}$$

while the standard deviations $\sigma_y$ and $\sigma_z$ as

$$\sigma_y(x) = k_w(x - x_0) + \sigma_{y0} \ , \tag{A10}$$

$$\sigma_z(x) = k_w(x - x_0) + \sigma_{z0} \ . \tag{A11}$$

All quantities with the subscript "0" represent wake properties at the far wake onset (the end of the near wake) and depend on turbine thrust coefficient $C_T$ and turbulence intensity $I$ (Eq. (A18)). Near wake length is computed with

$$x_0 = \frac{D\cos\gamma\left(1 + \sqrt{1 - C_T}\right)}{\sqrt{2}\left(4\alpha I + 2\beta\left(1 - \sqrt{1 - C_T}\right)\right)} \ , \tag{A12}$$

where $\alpha$ and $\beta$ are the tuning parameters governing the influence of turbulence and thrust coefficient on the near wake end location, respectively. Far wake onset wake widths in the vertical ($\sigma_{z0}$) and cross-stream ($\sigma_{y0}$) directions are defined as

$$\sigma_{z0} = 0.5D\sqrt{\frac{u_R}{U_\infty + u_0}} \ , \tag{A13}$$

$$\sigma_{y0} = \sigma_{z0}\cos\gamma \ , \tag{A14}$$

with the velocity at the turbine rotor $u_R$ and the velocity in the near wake $u_0$ (assumed as constant) computed with

$$u_R = \frac{U_\infty C_T}{2\left(1 - \sqrt{1 - C_T}\right)} \ , \tag{A15}$$

$$u_0 = U_\infty \sqrt{1 - C_T} \ . \tag{A16}$$

For additional details on the derivations of far wake onset quantities refer to Bastankhah and Porté-Agel (2016).

In the Gaussian model formulation, $k_w$ represents the main parameter. It defines the rate of linear wake expansion, and it accounts for ambient and added turbulence intensities. It is expressed by

$$k_w = k_a I + k_b \tag{A17}$$

where $k_a$ and $k_b$ are tuning parameters defining the weight of turbulence intensity and fundamental wake recovery, and $I$ represents the turbulence intensity, expressed as

$$I = \sqrt{\sum_{j=0}^{N}\left(I_j^+\right)^2 + I_0^2} \ . \tag{A18}$$

$N$ is the total number of turbines influencing the reference turbine, and $I$ considers both $I_0$, the ambient turbulence intensity, and $I_j^+$, the added turbulence intensity due to an upstream turbine $j$. Given an overlap area $A_{overlap}$ between the wake of an upstream turbine $j$ and the turbine of interest, $I_j^+$ is defined by the empirical expression by Crespo et al. (1999)

$$I_j^+ = A_{\text{overlap}} \left( 0.5 a_j^{0.8} I_0^{0.1} (x/D_j)^{-0.32} \right) , \tag{A19}$$

where the coefficients are tuned with LES simulations by King et al. (2021). For additional details on added turbulence calculations, refer to Niayifar and Porté-Agel (2016).

Although the wake model complexity is increased, the Gaussian wake model is still suitable for control and optimisation applications. Additional considerations include ambient and added turbulence intensity, and the wake model explicitly conserves momentum. The main limitations are the formulation based on a free shear approximation of the Navier-Stokes equations and the inaccurate near wake predictions. Moreover, it does not compute cross-wise and streamwise velocity components, critical for modelling wake steering effects, and it relies on multiple empirical coefficients, decreasing the range of conditions for suitable predictions.

### A1.4 Gauss-Curl Hybrid Model

The Gauss-Curl Hybrid (GCH) model (King et al., 2021) extends the modelling capabilities of the Gaussian model by implicitly modelling wake rotation and counter-rotating vortices due to yaw misalignment, allowing the description of wake asymmetry, added yaw-based wake recovery, and secondary-steering effects. The Gaussian model formulation is improved by introducing analytical approximations based on the Curl model (Martínez-Tossas et al., 2019), a recently developed linearised RANS formulation with an increased computational complexity of around 1000x compared to the Gaussian model. Further developments of the Curl model are included in Zong and Porté-Agel (2021). This section provides an overview of the GCH wake model. For additional details on the calculations, refer to King et al. (2021).

Wake rotation is included by modelling a Lamb-Oseen vortex, where its circulation strength is dependent on the turbine axial induction factor and tip-speed ratio. The counter-rotating vortex system due to turbine yaw misalignment is modelled as two single vortices released at the top and the bottom of the rotor, with a vortex strength dependent on turbine thrust coefficient and yaw angle. For each vortex described, cross-stream $V$ and vertical $W$ velocity components are calculated and linearly combined. Ground effects are considered by adding mirrored vortices below the ground. Vortices combination of the velocity components can be expressed by

$$V_{\text{wake}}(x, y, z) = V_{\text{top}} + V_{\text{bottom}} + V_{\text{wake rotation}} ,$$

$$\tag{A20}$$

$$W_{\text{wake}}(x, y, z) = W_{\text{top}} + W_{\text{bottom}} + W_{\text{wake rotation}} .$$

Vortices dissipation with downstream distance is computed as

$$V(x,y,z) = V_{\text{wake}}(x,y,z) \left( \frac{\epsilon^2}{4\nu_T \frac{(x-x_0)}{U_\infty} + \epsilon^2} \right) ,$$

(A21)

$$W(x,y,z) = W_{\text{wake}}(x,y,z) \left( \frac{\epsilon^2}{4\nu_T \frac{(x-x_0)}{U_\infty} + \epsilon^2} \right) ,$$

where $\epsilon$ is the vortex core size ($\epsilon = 0.2D$ in this study), and $\nu_T$ is the turbulent viscosity, defined with a mixing length model.

As presented in King et al. (2021), added wake recovery due to turbine yaw misalignment is considered by updating turbulence intensity with an additional component $I_{\text{mix}}$ due to yaw enhanced mixing. The updated turbulence intensity $I_{\text{updated}}$ is expressed as

$$I_{\text{updated}} = I + \phi I_{\text{mix}} ,$$

(A22)

where the tuning parameter $\phi$ is introduced due to the analytical approximations at the base of $I_{\text{mix}}$ calculations. $I_{\text{updated}}$ directly affects the wake expansion as specified in Eq. (A17) (Appendix-A1.3). $I_{\text{mix}}$ is calculated with

$$I_{mix} = \frac{\sqrt{\frac{2}{3} K_{e,\text{tot}}}}{\overline{U}} - I ,$$

(A23)

with $K_{e,\text{tot}}$ representing the total turbulent kinetic energy including yaw effects. Using approximate solutions for velocity fluctuations in three dimensions, $K_{e,\text{tot}}$ can be expressed by

$$K_{e,\text{tot}} = 0.5 \left( u'^2 + (v' + v_{\text{curl}})^2 + (w' + w_{\text{curl}})^2 \right) .$$

(A24)

Cross-stream and vertical velocity fluctuations are defined as the sum of the average cross-stream $v'$ and vertical $w'$ velocity contributions at the turbine and the upstream turbine contributions $v_{\text{curl}}$ and $w_{\text{curl}}$ from Eq. (A21). Streamwise velocity fluctuations $u'$ are derived from the turbulent kinetic energy $K_e$, based on the ambient turbulence intensity, and are defined as

$$u' = \sqrt{2K_e} , \quad \text{where} \quad K_e = \frac{3}{2} \left( \overline{U} I \right)^2 .$$

(A25)

Considering yawed upstream turbines, secondary steering refers to the phenomenon of downstream turbines without yaw misalignment experiencing wake deflection and deformation due to the interaction between upstream-generated vortices and downstream turbine wake rotation (see Fig. 1-d). The GCH wake model captures secondary steering effects by computing an effective yaw angle $\gamma_{\text{eff}}$, the yaw misalignment that a downstream turbine would require to produce the same cross-stream

velocity component $V_{\text{eff}}$ as the average one calculated at its rotor due to upstream vortices $\overline{V}$. $\gamma_{\text{eff}}$ is computed as

$$\gamma_{\text{eff}} = \text{argmin} |\overline{V} - V_{\text{eff}}| , \quad \text{where} \quad V_{\text{eff}} = V_{\text{wake}}(\gamma) .$$

(A26)

Defining $\gamma_{\text{turb}}$ as the rotor misalignment to the wind, the total yaw angle $\gamma$ at the turbine rotor, consequently used in the wake deflection calculations (Appendix-A1.3), is expressed as

$$\gamma = \gamma_{\text{turb}} + \gamma_{\text{eff}} .$$

(A27)

Although cross-wise and vertical velocity components are not explicitly modelled, the GCH wake model is capable of capturing, through analytical approximations, the increase in wake recovery due to yaw misalignment and secondary steering effects. The latter is particularly important when considering large wind farms and evaluating wake steering control strategies. However, GCH added complexity results in 3.5x higher computational requirements. Moreover, the parameter dependency for the Gaussian model still persists.

## A2 Deflection models

When a turbine is yawed, the unbalance of thrust on the rotor leads to a cross-stream momentum gain and a deflection of the wake in the direction of yaw. Deflection models aim at quantifying this shift on the streamwise distance $x$ and applying it to the streamwise velocity deficit field. All analytical models investigated in this study describe the velocity vector in one dimension only, leading to yaw solely affecting the streamwise velocity. Estimations of cross-stream and vertical velocity components are included in the GCH wake model (Appendix-A1.4), where an additional correction to the streamwise velocity field is applied based on these empirical estimations.

### A2.1 Jimenez Model

The Jimenez deflection model is based on the empirical formulation proposed by Jiménez et al. (2009). The two main assumptions are: first, the streamwise velocity deficit values are almost negligible compared to inflow velocity $U_\infty - \delta u \simeq U_\infty$, second, the wake skew angle $\zeta$ is small enough that $\cos(\zeta) \simeq 1$ and $\sin(\zeta) \simeq \zeta$. Considering a turbine offset to the freestream by a yaw angle $\gamma$, the wake angle at the centreline is defined as

$$\zeta(x, \gamma, C_\mathrm{T}) \approx \frac{(\zeta_\mathrm{init}(C_\mathrm{T}, \gamma))^2}{1 + 2k_\mathrm{d}\frac{x}{D}} \; ,$$

$$\zeta_\mathrm{init}(\gamma, C_\mathrm{T}) = 0.5 \cos^2\gamma \sin\gamma \; C_\mathrm{T} \; ,$$

(A28)

where $k_d$ is a tuneable deflection parameter. The amount of wake deflection $\delta(x)$ can be calculated by skew angle integration. Ambient turbulence intensity can be indirectly taken into account through the $k_d$ parameter. However, no considerations for added turbulence intensity and wake rotation are provided. Moreover, wake meandering is not accounted for due to the model's steady-state description.

### A2.2 Bastankhah Model

The Bastankhah deflection model is based on the budget analysis of the continuity and Reynolds-averaged Navier-Stokes equations conducted by Bastankhah and Porté-Agel (2016). Using vortex theory, the wake deflection angle at the rotor is defined by

$$\theta \approx \frac{0.3\gamma}{\cos\gamma} \left(1 - \sqrt{1 - C_\mathrm{T}\cos\gamma}\right) \; .$$

(A29)

By assuming the validity of Eq. (A29) is not limited to the calculation of flow angle at the rotor but it extends to the near wake region, and by approximating the near wake deflection to a constant value, the far wake onset wake deflection $\delta_0$ can be

expressed as

$$\delta_0 = x_0 \tan\theta\,, \tag{A30}$$

where $x_0$ is defined in Appendix-A1.3. The total far wake deflection due to wake steering is defined as

$$r\frac{\delta(x)}{D} = \frac{\delta_0}{D} + \frac{\theta}{14.7}\sqrt{\frac{\cos\gamma}{k_w^2 C_T}}\left(2.9 + 1.3\sqrt{1-C_T} - C_T\right)\ln\left[\frac{\left(1.6+\sqrt{C_T}\right)\left(1.6\sqrt{\frac{8\sigma_y\sigma_z}{D^2\cos\gamma}} - \sqrt{C_T}\right)}{\left(1.6-\sqrt{C_T}\right)\left(1.6\sqrt{\frac{8\sigma_y\sigma_z}{D^2\cos\gamma}} + \sqrt{C_T}\right)}\right]\,, \tag{A31}$$

with the main affecting factors being turbine operating conditions and ambient and added turbulence intensity.

Expressions for the symbols in the above equation are provided in Appendix-A1.3. Refer to Bastankhah and Porté-Agel (2016) for details on the model derivation. The Bastankhah deflection model does not include considerations about asymmetry due to wake rotation, calculations of vertical and cross-stream velocity components, or the effect of wake meandering due to the steady nature of the model predictions.

## Appendix B

In this appendix, the validation of the presented optimisation framework is provided. The optimisation problem in Rak and Santos Pereira (2022) is replicated, and results for Jensen, Gaussian and GCH wake models are compared. A single column of aligned NREL 5 MW turbines is optimised for farm power maximisation. The yaw angle of each turbine in the farm layout is set as a variable bounded by $[-50\,°, +50\,°]$. SLSQP optimisation algorithm by Kraft (1988) is used. Moreover, optimiser parameters and initial yaw settings are matched to the values provided in Rak and Santos Pereira (2022). The farm layouts and atmospheric conditions optimised are shown in Table B1. Results are provided in Fig. B1. Overall, a satisfactory agreement is reached. The trends for the optimal variable solutions are captured for all wake models. Moreover, a close match for optimal yaw angles can be observed, except for a few isolated conditions. These marginal discrepancies are attributed to some of the wake model parameters not being specified in Rak and Santos Pereira (2022) and to the use of different FLORIS versions (2.4 in the current work, 2.2 in Rak and Santos Pereira (2022)).

**Table B1.** Investigated wind farm layouts and atmospheric conditions from Rak and Santos Pereira (2022).

| Case name | ws [m s$^{-1}$] | wd [$deg$] | TI$_0$ [%] | spacing [D] |
|---|---|---|---|---|
| Reference case (RC) | 8.0 | 270.0 | 7.5 | 7.0 |
| High wind speed (HWS) | 13.0 | 270.0 | 7.5 | 7.0 |
| Wind direction 275 $°$ (WD 275) | 8.0 | 275.0 | 7.5 | 7.0 |
| Low turbulence intensity (LTI) | 8.0 | 270.0 | 5.0 | 7.0 |
| Small spacing (SS) | 8.0 | 270.0 | 7.5 | 5.0 |

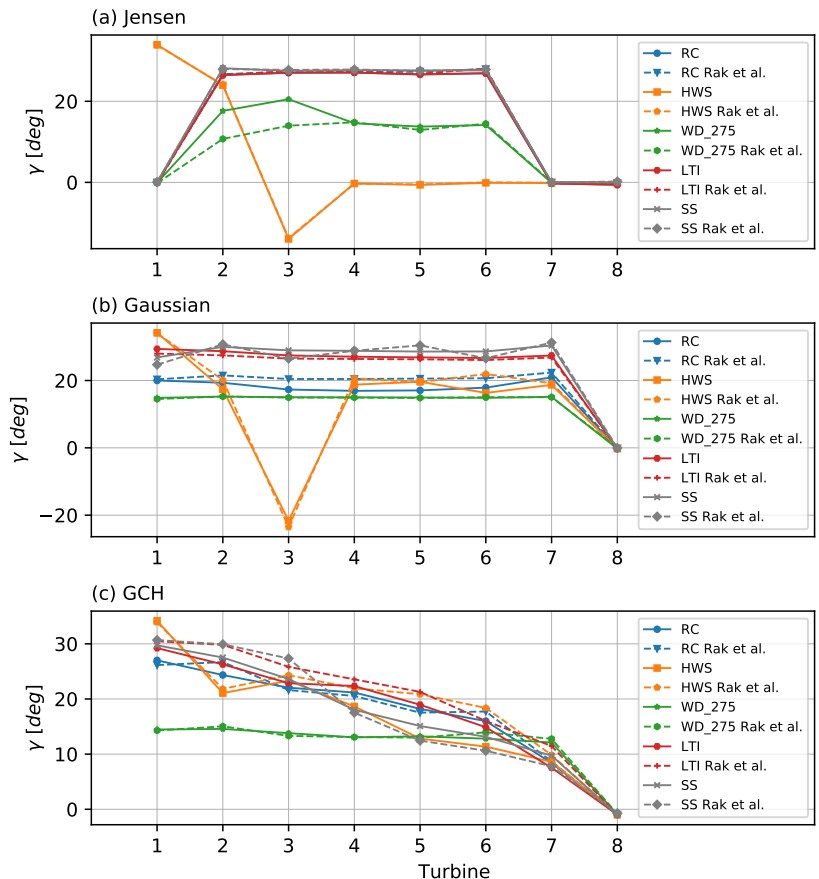

**Figure B1.** Validation of the presented optimisation framework. Results are shown for the optimal turbine yaw angle comparison with Rak and Santos Pereira (2022) for (**a**) Jensen, (**b**) Gaussian, and (**c**) GCH wake models.

### Appendix C

This appendix shows an illustrative example of a simple 1-D minimisation problem with Bayesian optimisation. The presented approach does not specifically refer to the TuRBO algorithm (described in Sect. 3.1) but rather aims to provide the reader with the general concepts and terminology of Bayesian optimisation. Before analysing the specific minimisation problem, a summary of the general idea behind Bayesian optimisation is provided. First, an initial sampling of the design space is performed (e.g. using Latin Hypercubes, Sobolo sequences, Hammersley sequences, or Monte-Carlo sequences). Second, the

true objective function to be minimised is approximated with a surrogate model, usually a Gaussian process. This probabilistic model provides a posterior distribution with a mean and variance of the surrogate model. Next, a new sample point at which to evaluate the underlying objective function is computed. This is achieved by minimising a criterion specified through an acquisition function. Commonly used acquisition functions include probability of improvement (PI), expected improvement

(EI), Thompson sampling (TS), and upper confidence bound (UCB). As more samples are evaluated, the surrogate model aims
to improve its representation of the true function until meeting a specified convergence criterion.

As an indicative 1-D example of a multi-modal objective function, let us consider the problem of minimising

$$f(x) = -(1.4 - 3x)\sin(18x), \qquad 0.4 \leq x \leq 1.1.$$

Figure C1 shows the development of the Gaussian process posterior over eight iterations of a typical Bayesian optimisation
implementation using an Expected Improvement (EI) acquisition function. Each sub-figure includes the evaluated points (black
filled circles), the posterior mean (black line) and uncertainty (blue shaded area), the true function (dashed green line), the
acquisition function (red line), and the location of the next evaluation point (red dashed vertical line). The initial design space
sampling is shown in Fig. C1-a and consists of three randomly-located points close to a local minimum. In Fig C1-b,g, the EI
acquisition function selects new evaluation points, progressively reducing the posterior uncertainty and improving the mean
prediction of the true function. Due to the EI function balancing exploitation and exploration, the newly selected points focus
on evaluating the minimum of the posterior mean and also on exploring areas with high uncertainty. Finally, the minimum of
the posterior mean approximately coincides with the true global minimum in Fig. C1-h.

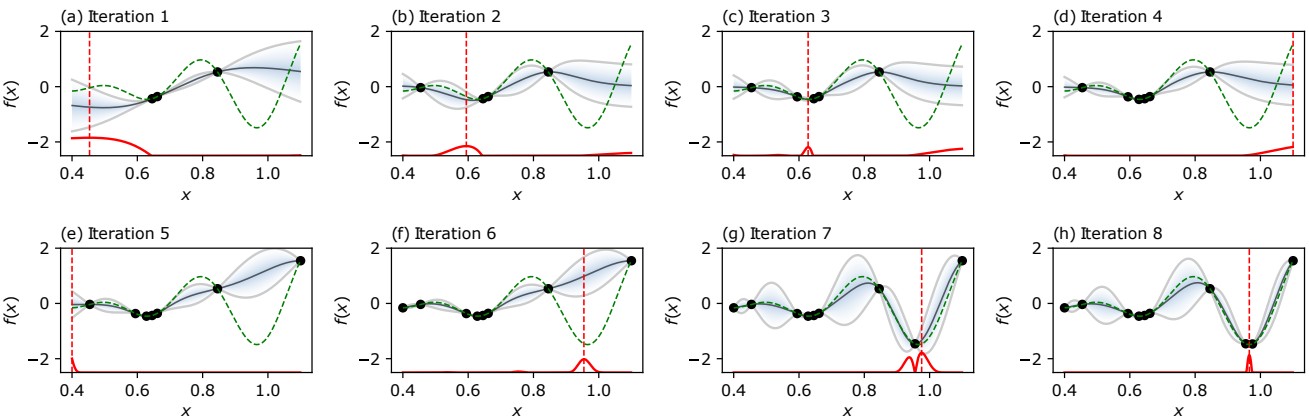

**Figure C1.** Illustrative example of a 1-D minimisation problem using Bayesian optimisation. Each subplot refers to an iteration of the
optimisation algorithm and includes the evaluated points (black filled circles), the posterior mean (black line) and uncertainty (blue shaded
area), the true function (dashed green line), the acquisition function (red line), and the location of the next evaluation point (red dashed line).

*Author contributions.* F. Gori: optimisation computations, Methodology, Validation, Data Analysis, Writing. A. Wynn: Methodology, Data
Analysis, Funding acquisition, Supervision, Writing. S. Laizet: Methodology, Data Analysis, Funding acquisition, Supervision, Writing.

*Competing interests.* The authors declare that they have no conflict of interest.

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
