# Peer review of "Sensitivity analysis of wake steering optimisation for wind farm power maximisation"

_Wind Energy Science, 2023_

## Referee Comment (RC1)

[referee-annotated manuscript omitted]

---

## Author Comment (AC1)

**Authors' response to Review 1 of "Sensitivity analysis of wake steering optimisation for wind farm power maximisation"**

F. Gori, S. Laizet, A. Wynn

April, 2023

**Main comment**

The authors would like to thank the reviewer for the in-depth review of the paper. The authors' responses (in blue) to all major and moderate issues (in black) are included in this document. Minor issues have been addressed with appropriate modifications to the text. All changes are marked in red in the revised manuscript.

**Major concerns:**

1. I am considerably concerned with the fact that the authors claim [line 135] to be using the exponential loss of power according to the yaw angle as $w = 0.627$ following Fleming et al., (2017), whereas $w$ (pP therein) is therein 1.43 or 1.88 depending on the chosen turbine, and 1.41 when fitted into a cosine function. They need to justify the use of the chosen value ($w = 0.627$) in order to being able to proceed with the paper, otherwise it would be a rejection as this parameter is applied in all the analytical models used (it is embedded in the FLORIS framework). In addition, you are applying the same $w$ value for the different turbines considered, whereas Fleming et al. (2107) evidences a different value when considering different turbines. The validity of this parameter must be checked before proceeding with the rest of the review, because if its invalidity was verified the whole work would be needed to be re-done (thus rejected herein).

   The authors can reassure the reviewer that there is no issue with the $w$ parameter in the manuscript. According to Equation 3, the correction to power due to yaw misalignment is $((\cos\gamma)^w)^3$, with $w = 0.627$, and

   $$((\cos\gamma)^w)^3 = (\cos\gamma)^{w\times 3} = (\cos\gamma)^{pP}.$$

   Hence, the parameter value of $w = 0.627$ used in the manuscript matches the coefficient $pP = 3 \times 0.627 = 1.881$ recommended in Fleming et al. [2017].

   Regarding the reviewer's comment on using the same $w$ value for the NREL 5 MW and the Vestas V-80 2 MW, the authors can confirm that the value of $w = 0.627$ (tuned value for the NREL 5 MW presented in Fleming et al. [2017]) has been also used for the Vestas V-80 2 MW. This decision was made after investigating the influence of $w$ on an SLSQP wake steering optimisation of the Horns Rev wind farm layout (80 Vestas V-80 2 MW) with the GCH wake

[Figure]

Figure 1: Comparison of optimisation results for different $w$ coefficient values. The farm layout is representative of the Horns Rev wind farm, the optimisation algorithm is SLSQP, and the wake model used is the GCH. (a): optimal set-points per turbine. Dashed lines delimit wind farm rows. (b): resulting objective function (normalised farm power).

model. As it can be seen in Figure 1, the results indicate that the largest variation is 7 % in power improvement and 5 ° in optimal yaw settings for a single turbine. These differences in optimisation results do not fundamentally affect the main trend in the obtained optimal solution (i.e. a monotonic decrease of optimal yaw angles with increasing downstream turbine location) and show that the conclusions of the original manuscript are not affected by the choice of $w$. It should be noted that there is a lack of power data for the Vestas V-80 2 MW turbine in yawed conditions. As an example, no specific value of $w$ for the Vestas V-80 2 MW is stated in [Dou et al., 2020]. The manuscript has been modified to include the discussion on the validity of $w$ in the presented study.

2. In sections 4.2.1 and 4.2.2 the authors show several times that, the used optimization algorithm (SLSQP) has serious problems to find global maxima for each turbine individually and therefore for the overall wind farm. Then in Section 4.2.3 they show the results statistics or TuRBO and SLSQP, showing a clear outperforming by TuRBO, a stochastic optimization approach. Indeed, the fact that stochastic algorithms present higher skill than gradient-based algorithms is shown in contributions such as e.g, see Kuo et al., 2020: "Wind Farm Yaw Optimization via Random Search Algorithm". For both reasons, it is worrying surprising that they do not apply TuRBO for the analysis of the long sections 4.2.1 and 4.2.2. Please re-do 4.2.1 and 4.2.2 sections (at least also) for the TuRBO algorithm, or consider dismissing such sections.

The purpose of Sections 4.2.1 and 4.2.2 is to highlight not only that unconstrained gradient-based optimisation can perform poorly, but also to explain why this happens. The observations made in this section (i.e., initial condition sensitivity and cost function bimodality) then motivate the proposed solutions to this problem via global optimisation (TuRBO) or constrained gradient-based optimisation in Section 4.3. For this reason, the authors believe that the insights gained in Sections 4.2.1 and 4.2.2 are necessary to motivate the paper.

To address the reviewer's comment, the authors have:

1. Re-run the optimisation presented in Figure 5 to include the output of the TuRBO algorithm. These new results are discussed at the end of section 4.2.1 and now provide further motivation for the statistical comparison of TuRBO and SLSQP in Section 4.3.

2. Replaced the ten indicative SLSQP optimization runs from Figure 4 with an analogous sample of ten optimisation runs selected from the 50 cases studied in Section 4.3. Consequently, both TuRBO and SLSQP have been applied to every initialisation case considered in Figures 4,5 and 8. For legibility, TuRBO's output for the ten cases of Figure 4 is not shown, but note that the improved performance of TuRBO can now be inferred from the statistical results presented in Figure 8.

3. Added more discussion to Sections 4.2 and 4.3 comparing the pros and cons of stochastic algorithms versus constrained/unconstrained gradient-based algorithms.

Finally, the purpose of Section 4.2.2 is only to discuss the geometry of the underlying objective functions for each wake model and use any observations to understand the poor performance of gradient-based wake steering optimisation. No optimisation is performed in this section, meaning that it is not possible to generate extra results here using TuRBO.

3. With their statement at Line 370: "To the best of the authors' knowledge, the work by Dou et al. (2020) is the only published study in which wake steering optimisation is performed on the Horns Rev farm layout" the authors seem not to handle the literature in depth, as they fail to raise other contributions such as the one by Zong and Porté-Agel (2021): 'Experimental investigation and analytical modelling of active yaw control for wind farm power optimization' (Section 6). This will entail (at least) serious implications in their discussion on their obtained results.

The authors thank the reviewer for raising this additional work, which is valuable to include. Nevertheless, it is clear that only a few studies on wake steering optimisation for the Horns Rev farm layout have been published to date, meaning that it does not have serious implications for the manuscript discussion. This study by Zong and Porté-Agel [2021], which uses gradient-based optimisation, requires explicit knowledge of the optimal solution to define appropriate initial yaw angles which guarantee optimiser convergence. This is a strong limitation to which our work offers a simpler alternative based on constrained gradient-based optimisation (Section 4.3). The revised manuscript has been modified accordingly to reflect this comment from the reviewer.

4. As a follow-up from point 3, a flagrant lack of state of the art on wake steering optimization contributions (at least those on open-loop) is noted in the introduction.

The authors' intention in the original submission was to present a concise overview of the literature on wake steering optimisation, as they wanted the focus of the paper to be on the results and new optimisation approaches proposed. To address the reviewer's comment, the introduction in the revised manuscript has been expanded to include a wider discussion of the literature on wake steering optimisation.

5. Numerous descriptive elements of the introduction and methodology are misplaced (those at the introduction should be at the methodology and vice-versa), and the paper shows in general serious lack of appropriate writing style. All my comments regarding these two aspects are indicated in the attached PDF.

The authors appreciate the reviewer's comment even if they disagree with the use of the word "serious" to describe their writing style. Numerous appropriate modifications have been made in the revised manuscript following this comment in an attempt to improve the quality of the study.

6. Finally, due to the fact that the analytical models used were not validated against ground truth (field campaigns) or more robust representations (e.g., wind tunnel experiments, high resolution numerical simulations as LES), it is very difficult to detect which analytical model is actually performing better. Therefore I would at least recommend to apply the optimized yaw angles obtained with a given model into the rest of models (inter-model cross-validation), and check how solutions perform.

As stated in Section 2, lines 106–107 of the original manuscript, the study does not seek to identify which wake model is best for wake steering optimisation in terms of modelling accuracy. Much past literature, such as [Göçmen et al., 2022, King et al., 2022], addresses this research question. The purpose of this submission study is instead to identify and analyse situations in which wake steering optimisation is highly sensitive to either wake model choice or the optimisation strategy used.

Regarding cross-validation, Table 1 shows, for the Horns Rev example of Section 4.3, the farm power increase for the Jensen, Multizone and Gaussian models when (i) using optimal decision variables computed using the GCH model and (ii) using the optimal decision variables from the original submission. For all three models, lower farm power increases are achieved using the optimal CGH yaw angles. This implies that optimising using the Jensen, Multizone, and Gaussian models will not give the characteristic row-monotonic behaviour of the optimal GCH yaw angles.

As noted in [Zong and Porté-Agel, 2021], this row-by-row decrease in optimal yaw angles arises due to secondary-steering effects, which are captured in the GCH model but are not in the Gaussian model. It is therefore arguable that optimisation using the GCH model exhibits the best performance out of all models considered.

Some text has been added in the revised manuscript to discuss inter-model cross-validation and which wake model is potentially performing better.

| Case | Jensen $P_{wf}P(0)^{-1}$ | Multizone $P_{wf}P(0)^{-1}$ | Gaussian $P_{wf}P(0)^{-1}$ |
|---|---|---|---|
| Original model $\gamma_{opt}$ | 1.04 | 1.83 | 1.12 |
| GCH $\gamma_{opt}$ | 1.03 | 1.39 | 1.07 |

Table 1: Inter-model cross-validation for C1+C2 optimisation results of the Horns Rev case.

**Moderate concerns (orange comments extracted from the reviewer's PDF file):**

1. L146: *"80 Vestas V-80 2 MW turbines"*.

   Reviewer's comment: You apply the same w value to different turbines, even though in Fleming et al., (2017) a considerable w (pP therein) difference is found among different turbines. Please justify.

   Please refer to authors' reply for major concern 1.

2. L173-L174: *"Moreover, a single trust region is used in all cases to enable a fair comparison with the SLSQP algorithm"*

   Reviewer's comment: Could you please explain why a fair comparison is attained by only considering ONE of the trust regions at TuRBO?

If multiple trust regions are used, then TuRBO uses a multiple starting point search and fits a different Gaussian process model in each trust region. Hence, a single run of TuRBO would be comparable, in complexity, to multiple runs of SLSQP. Therefore, the simplest fair comparison between the two methods is to use a single trust region for TuRBO, initialised with the same conditions as those used for each SLSQP optimisation. To address this comment, a clarification has been added to Section 3.1 of the revised manuscript.

3. L335-L336: *"Higher mean values are obtained for all models by the global TuRBO algorithm."*

   Reviewer's comment: If you want to keep sections 4.2.1 and 4.2.2 you must re-do them in terms of (at least also) TuRBO. It does not make any sense to show them in terms of just SLSQP when they are performing that poorly, especially when you are showing now these comparisons with an outperforming aproach such as TuRBO.

   Please refer to authors' reply for major concern 2.

4. L348-L352: *"It is interesting that the best-performing SLSQP run (blue markers) converges after fewer iterations, and to a higher farm power, than the indicative TuRBO case presented and, in fact, outperforms all TuRBO runs (see Figure 8). This suggests that, if initialised in the region of attraction of a global (or near-global) maxima, a gradient-based optimiser may exhibit faster convergence than a generic global strategy."*

   L356-L361: *"Finally, it is interesting to note that while the optimal wake velocity fields in Figures 9-b and 9-c are visually similar, they are not identical: the TuRBO solution has 1.5 % lower farm power, and requires 55 % more evaluations to find it. This confirms a possible, albeit counter-intuitive, advantage of the rapid local convergence enjoyed by gradient-based optimisers. Indeed, the quasi-Newton algorithm employed in SLSQP is well-known (Deuflhard, 2011) to possess rapid locally-quadratic convergence rates, which may partially explain this observation."*

   Reviewer's comment: However, due to the high variability of runs in results not validated against ground truth (field data) or at least more robust reproductions (wind tunnel, high resol. simulations such as LES), it is very hard to justify an option with such big dispersion as with SLSQP. In absence of such validation, it would be preferable to trust the optimizer providing more reliability in terms of least dispersed results and higher average performance (i.e., TuRBO). Please justify more robustly or remove.

   A clarification has been added to Section 4.2.3 to emphasise that unconstrained optimisation using SLSQP gives more variable results than TuRBO and, consequently, that TuRBO should be viewed as a more robust optimiser than unconstrained SLSQP.

   Text has also been added to emphasise that: (i) SLSQP's variability is precisely the motivation for considering constrained optimisation in Section 4.3; and (ii) that, as shown in Figure 11, extra constraints can significantly reduce the variability of SLSQP. Consequently, the authors argue that appropriately constrained gradient-based optimisation is also a viable solution to robust wake steering optimisation.

5. L369-L370: *"To the best of the authors' knowledge, the work by Dou et al. (2020) is the only published study in which wake steering optimisation is performed on the Horns Rev farm layout."*

   Reviewer's comment: The authors knowledge is incomplete. Please check Zong and Porté-Agel (2021): 'Experimental investigation and analytical modelling of active yaw control for wind farm power optimization' (Section 6).

Please refer to authors' reply for major concern 3.

**Minor concerns (yellow comments in the reviewer's PDF file):**

These comments are addressed directly in the revised manuscript (red colour in the text). The authors have also dealt with the recommendations from the referee to remove small pieces of text when they felt that it was appropriate to do so.

**References**

P. Fleming, J. Annoni, Jigar J. Shah, L. Wang, S. Ananthan, Zhijun Zhang, Kyle Hutchings, Peng Wang, Weiguo Chen, and Lin Chen. Field test of wake steering at an offshore wind farm. *Wind Energy Science*, 2(1):229–239, 2017.

B. Dou, T. Qu, L. Lei, and P. Zeng. Optimization of wind turbine yaw angles in a wind farm using a three-dimensional yawed wake model. *Energy*, 209, October 2020. ISSN 03605442.

H. Zong and F. Porté-Agel. Experimental investigation and analytical modelling of active yaw control for wind farm power optimization. *Renewable Energy*, 170:1228–1244, June 2021. ISSN 0960-1481. doi: 10.1016/j.renene.2021.02.059.

T. Göçmen, F. Campagnolo, T. Duc, I. Eguinoa, S. Andersen, Vlaho Petrović, Lejla Imširović, Robert Braunbehrens, Jaime Liew, Mads Baungaard, Maarten Laan, Guowei Qian, Maria Aparicio-Sanchez, Rubén González-Lope, Vinit Dighe, Marcus Becker, Maarten Broek, J. W. Wingerden, Adam Stock, and Johan Meyers. FarmConners wind farm flow control benchmark – Part 1: Blind test results. *Wind Energy Science*, 7:1791–1825, September 2022.

J. King, P. Fleming, L. Martinez, C. Bay, and M. Churchfield. Aerodynamics of Wake Steering. In Bernhard Stoevesandt, Gerard Schepers, Peter Fuglsang, and Yuping Sun, editors, *Handbook of Wind Energy Aerodynamics*, pages 1197–1221. Springer International Publishing, Cham, 2022. ISBN 978-3-030-31307-4.

---

## Author Comment (AC2)

**Authors' response to Review 2 of "Sensitivity analysis of wake steering optimisation for wind farm power maximisation"**

F. Gori, S. Laizet, A. Wynn

May, 2023

**Main comment**

The authors would like to thank the reviewer for the careful reading of the paper and the in-depth review. The authors' responses (in blue) to all reviewer's comments (in black) are included in this document. All changes are marked in red in the revised manuscript.

**General:**

1. I'm not sure the inter-model comparisons make a good contribution to the paper. The different models give different results, but often it should be noted that this is the intention, in other words, the reason model development continued was to improve the match to higher-fidelity models of cases similar to the ones explored in this paper. The differences are therefore somewhat the point of the model, and so relate a bit awkwardly to the paper's framing of sensitivity.

   Our motivation for using inter-model comparisons has been clarified with an addition to the second paragraph of Section 2.

   For completeness, we have two reasons for studying inter-model behaviour. First, if a model is used in wake steering optimisation, it is important to ask how complex the model should be to produce broadly the correct decision variables: lower fidelity models may still perform well in an optimisation context. Second, even if two models give similar optimal variables, it is important to know which model gives good solutions more consistently (e.g. under different initialisations).

   The examples given in our submission help clarify these points. First, while the Gaussian and Multizone models can broadly give the same optimal yaw angles, the Gaussian model is much more robust (see Figure 8-b,c). Second, while the GCH model provides more physically accurate optimal yaw angles compared to the Gaussian model (due to secondary steering effects), GCH optimisation results are more variable (Figure 8-c,d), and hence less robust.

   We do not believe that, to-date, the question finding the "correct" model fidelity for wake steering optimisation has been fully resolved. Our intention is that the presented inter-model comparison makes a step towards answering it.

2. I thought the comparisons between optimization strategies were useful and interesting though. However, the description of the turbo algorithm was a little too brief for me to fully understand its approach (I'm also not familiar with several of the terms).

The description of the TuRBO algorithm in Section 3.1 has been extended to provide further insight into Bayesian optimisation and the specifics of TuRBO. Moreover, an illustrative 1D minimisation problem on a toy function using a Bayesian optimisation has been added in Appendix-C.

3. I think in general the paper would benefit if some aspects were condensed and de-emphasized:

   (a) The Jensen model does a poor job at wake steer modeling, I don't think it was really designed for this so the time spent describing the model or looking at results of this model I think could be given over to better use

   We agree that our study shows that the Jensen model performs poorly in wake steering optimisation. However, the Jensen model remains widely used by the wind energy community for wake steering applications [Kheirabadi and Nagamune, 2019, Houck, 2021, Andersson et al., 2021]. For this reason, we believe it is valuable to demonstrate its high sensitivity for at least the simple and medium-complexity cases considered in this study (the $2 \times 1$ and $5 \times 5$ farms).

   For brevity, however, we have removed the Jensen model from the results for the Horns Rev example.

   (b) Then more generally, the difference of the results between models is also less interesting I think because the models were anyway designed to produce different results, so this does not need to be proved in my opinion.

   Please refer to answer to Comment 1, above.

4. It would be interesting to know more about the Turbo method, not just the general theoretical description of the method, but how specifically it is implemented in this case and why it is out-performing SLSQP

   Section 3.1 has been expanded to give a more detailed description of the TuRBO algorithm. In addition, a simple 1D example has been added to Appendix C to help visualise the iterative behaviour of Bayesian optimisation algorithms (of which TuRBO is an example).

   These added details help to explain the main differences between the two considered algorithms: at each iteration, TuRBO's global approach can potentially sample from anywhere in the design space, while SLSQP typically takes a small step computed using local gradients. This observation underpins the discussion of many of the presented results (e.g. the sensitivity to initialisation demonstrated in Sect 4.1).

5. Is this result specific to Turbo vs SLSQP, or would it be expected to generalize to similarly structured optimizers?

   Our results (see, e.g., Figures 4 and 6) demonstrate that the farm power improvement objective function typically has multiple local maxima and, for the Multizone and GCH models, has either discontinuities or discontinuous gradients. This behaviour implies that significant differences should be expected between the performance of any typical global optimiser (e.g., TuRBO) and the performance of any typical gradient-based optimiser.

6. A diagram of the TuRBO method would be welcome.

An illustration of Bayesian optimisation applied to a simple 1D minimisation problem has been added in Appendix-C. For further examples, please also see [Shahriari et al., 2016, Figure 1] and [Eriksson et al., 2019, Figure 1].

7. Would other optimizers compare interestingly? Genetic annealing? Serial-Refine?

   Consistent with the answer to Comment 6, above, we expect an improvement in performance for global strategies (e.g. genetic annealing) over local searches, which include gradient-based (e.g., SLSQP) or gradient-free (e.g., Serial-Refine) approaches, due to the multi-modal and discontinuous/non-smooth nature of the objective function.

8. In terms of optimization, layout optimization (or coupled layout/control design) is a harder problem for optimizers, since there are many more variables. Would turbo be interesting for those doing research in layout design?

   TuRBO would be useful in this context due to its ability to reduce optimisation sensitivity compared to unconstrained gradient-based optimisation (e.g. the recent use of Bayesian optimisation in farm layout optimisation [Bempedelis and Magri, 2023]). However, a known drawback of Bayesian approaches is the computational cost involved with tuning the Gaussian processes in high-dimensional optimisation problems which require many objective function evaluations. Although TuRBO mitigates this issue by using trust regions, it may also be promising to take inspiration from our findings and add physically-inspired constraints to enable efficient gradient-based algorithms for layout optimisation problems.

**Specific:**

9. Section 4: Does it make sense to include the Jensen model in these investigations, such as Figure 2? For instance you state:

   *"Model sensitivity is caused by the flatness of the Jensen model's downstream power curve apparent in Figure 2-b, that is, $\frac{d}{d\gamma_1}P_{T_2}(\gamma_1)$ is significantly smaller for Jensen than for other models. Flatness arises due to the uniform, or "top-hat", profile of the Jensen distribution (see Figure 1), which results in a lack of sensitivity 210 of streamwise velocity deficit to moderate yaw perturbations."*

   The inability of the flat Jensen distribution to model the impact of wake steering (or fit to LES data) was a motivation for the development of the multi-zone model (Gebraad, P. M. O., Teeuwisse, F. W., van Wingerden, J. W., Fleming, P. A., Ruben, S. D., Marden, J. R., and Pao, L. Y. (2016) Wind plant power optimization through yaw control using a parametric model for wake effects—a CFD simulation study. Wind Energ., 19: 95– 114 doi: 10.1002/we.1822.).

   Please refer to the answer to Comment 3, above.

10. Figure 4: This is an interesting result. But I might recommend you also check the slightly misaligned cases (for example 1 deg off in either direction) in addition to the aligned, these problems can be special to circumstances where wind turbine rows are perfectly aligned to the inflow, which is actually not the dominant case in a practical setting.

    The results and conclusions of Section 4.2 are also valid for slightly misaligned cases. Figure 1 of this document replicates the conditions of Figure 8 in the paper but extends the results

for slightly misaligned conditions. The objective function statistics are found to be consistent with those presented in our manuscript. Similar initialisation sensitivities, as well as model and optimiser dependencies, are observed for at least 2° of misalignment from fully aligned conditions. As the revised manuscript is already fairly long, we have decided not to add this figure in the revised manuscript and to only have a small comment for slightly misaligned cases.

[Figure]

Figure 1: Comparison of the $5 \times 5$ objective function statistics in wind direction range $[260, 270]$ between SLSQP (**a-d**) and TuRBO (**e-h**) optimisation algorithms for (**a,e**) Jensen, (**b,f**) Multizone, (**c,g**) Gaussian, and (**d,h**) GCH wake models. Mean values are illustrated with error bars for one standard deviation (red) and minimum and maximum values (black). Vertical dashed lines indicate fully aligned conditions, while horizontal dashed lines correspond to no power improvement.

11. Figure 7: Interesting find, is this still an issue in the latest versions of FLORIS?

    The results are independent of FLORIS, as long as the exact same wake, deflection and superposition models are used.

12. Section 4.3: The C2 constraint is a good idea, but is it complicated to carry out in all cases of wind directions? Can it be included in the optimization which turbines are upstream of which in every wind direction?

    The C2 constraint can be easily adapted to all cases of wind directions. For example, a simple permutation of the labelling of the turbines can be performed to obtain columns of turbines aligned with the downstream direction. The C2 constraint could then be applied in an equivalent manner to the one described in our study, using the new labelling.

    A small clarification has been added to the end of Section 4 to explain this point.

**References**

A. C. Kheirabadi and R. Nagamune. A quantitative review of wind farm control with the objective of wind farm power maximization. *Journal of Wind Engineering and Industrial Aerodynamics*, 192: 45–73, September 2019. ISSN 01676105.

D. R. Houck. Review of wake management techniques for wind turbines. *Wind Energy*, 25(2):195–220, 2021. ISSN 1099-1824.

L. E. Andersson, O. Anaya-Lara, J. O. Tande, K. O. Merz, and L. Imsland. Wind farm control - Part I: A review on control system concepts and structures. *IET Renewable Power Generation*, 2021. ISSN 17521424.

B. Shahriari, K. Swersky, Z. Wang, R. P. Adams, and N. de Freitas. Taking the Human Out of the Loop: A Review of Bayesian Optimization. *Proceedings of the IEEE*, 104(1):148–175, January 2016. ISSN 1558-2256.

D. Eriksson, M. Pearce, J. Gardner, R. D. Turner, and M. Poloczek. Scalable Global Optimization via Local Bayesian Optimization. In *Advances in Neural Information Processing Systems*, pages 1–12, December 2019.

N. Bempedelis and L. Magri. Bayesian optimization of the layout of wind farms with a high-fidelity surrogate model. 2023. doi: 10.48550/arXiv.2302.01071.

---

## Editor Decision (ED1)

Line numbers refer to the last version of the manuscript.

Minor changes:

- The use of the term "operating set-point" in some parts of the paper to refer to the layout or atmospheric conditions (e.g. in L58) could be misleading in a control context where set-point may refer to command. A rather preferred term could be for instance "operating conditions".

- Eq. (4). The total farm power (P) could be named as ($P_{WF}$) to better match the subsequent reference in Section 4. The subscript WF could also be applied to Eq.(5).

- L209, I'd suggest to include the explanation of the term multi-modality (L261: "the presence of multiple and distinct maxima") when the term is first mentioned in the text.

- Table 2. Maybe you could include somehow that the second range for eps corresponds to the C1 constraint, since this is not explained until L454. Reference to Table 2 could also be made in L456.

- Figure 2: The figure is not fully colour-blind friendly, specifically the lines corresponding to Gaussian and GCH models, which are very close, of similar line style and have the two most confused colours (green and red). Could you please kindly modify it to make it clearer for everyone?

- For the sake of clarity, please indicate in all figure captions to which layout corresponds the figure. Missing references in Fig. 2 (2x1 wind farm), Fig. 5-9 (5x5 wind farm) and Fig. 10-12 (Horns Rev wind farm).

- For the sake of higher colour-blind friendliness, in the discussion about Fig. 5, when referring to the markers, please include not only the colour (blue, green, red), but also the type of the marker (square, triangle, circle). Please also include the type of marker in the figure caption. The same goes for the discussion of Fig. 9.

- L295. Could you please provide any answer on RC1 comment about the statement "which saturates the upper bound constraints at 25º for all but the final turbine row", which refers the authors to previous results in the literature (Zong and Porté-Agel, 2020)? The comment is also related to those in Fig. 10 by RC1.

- L301. According to your answer to major concern 2 by RC1, you have selected the 10 random cases in Fig. 4 from the 50 cases studied in Section 4.3. It might be worth mentioning it in the section.

- L358. The comment refers to both Fig. 5-c and 5-g, so you could make reference to both sub-figures.

- L407. In the discussion about Figure 9, maybe the use of "the first case", "the second and third cases" is not the most adequate since this order does not match the order in the legend or in the velocity plots.

- L412, I guess you refer to Figure 8-d specifically.

- L416, in the reference to the figure you could also include "Fig. 9-d"

- L418, you could also include a reference to "Fig. 9-b"

- L427, in the reference to Fig. 5, please specify to which sub-figure you want to make reference, if any in particular.

- L454. When introducing constraint C1, could you please include an indication on why you choose the positive yaw angles as supposed to lead to more global maxima or to preferred solutions?

- L464. For the sake of clarity, please also include that the so-called nominal optimisation does not apply constraint C2 either, as RC1 suggested.

- L467, please make reference to Table 2.

- L476. To better complement the discussion and demonstrate the overall effect of the optimisation constraints, could you please include (e.g. in a table) the normalised farm power obtained by each of the optimisations depicted in Fig. 10? This might not be relevant in model comparisons for all the reasons discussed with referees, but can be of interest when analysing a particular model with different optimisation approaches. This may also help to differentiate the C1 and C1+C2 approaches. Do the results from the latter justify its greater complexity? Additionally, some comment about the number of runs required would be of interest, as RC1 commented.

- L504. Could you please better support the statement "For all wake models, the row-averaged probability density functions indicate an overall reduction in optimal yaw angle dependency to initial conditions for the constrained cases" by indicating from which specific aspect of the results this conclusion derives? (Comment by RC1).

- L506. Could you please specify in which subfigures it is observed the following statement: "Higher probability density function magnitudes can be observed, with more than double the values for some row distributions"? (Comment by RC1).

- There are several comments by RC1 pointing at the benefits that stochastic optimisation could also provide to the problem under study, referring to (Kuo et al., 2020). For the sake of completeness, the authors could include some comment (e.g. in the introduction) showing this as an alternative option, not covered in this manuscript but that could also be used to address the limitations of SLSQP approaches. It is true

that the reference indicated by RC1 is now included in the introduction but just as part of a mere listing of optimisation techniques without further indication. If you do not consider RC1 comments as pertinent, please provide some explanation of why.

- Some parts of the authors' answer to RC2 would be of interest in the text of the manuscript. For instance, aspects from the third paragraph of the answer to comment 1 could be nicely included in the discussion. Also, the main differences between SLSQP and TuRBO algorithms stated in your answer to comment 4.

- Please check and harmonise the references to figures throughout the text so that they fulfil the journal guidelines:
  The abbreviation "Fig." should be used when it appears in running text and should be followed by a number unless it comes at the beginning of a sentence, e.g.: "The results are depicted in Fig. 5. Figure 9 reveals that...".
  https://www.wind-energy-science.net/submission.html#figurestables

- According to the journal guidelines (https://www.wind-energy-science.net/submission.html#references) you will be requested to include a persistent identifier (DOI preferred) for all the references. So, you could already advance those modifications from now.

Typos:

- Figure 5: required space "the(a)"

- L508. Multizone and Gaussian nominal cases would be related to Fig. 12-a and 12-b instead of Fig. 12-b and 12-c.

- L512. Fig. 12-c instead of Fig. 12-d?

- L519. Please eliminate "a" in the sentence "This can achieved by a permuting the turbine…".

- L642. Subscript of the rotor velocity in the text would be with capital letter "R" to match the equations.

---

## Author Response (AR2)

**Authors' response to the editor's review of "Sensitivity analysis of wake steering optimisation for wind farm power maximisation"**

F. Gori, S. Laizet, A. Wynn

June, 2023

**Main comment**

The authors would like to thank the editor for the review of the paper. The authors' responses (in blue) to all comments (in black) are included in this document. Modifications are marked appropriately in the revised manuscript.

**Minor changes:**

1. The use of the term "operating set-point" in some parts of the paper to refer to the layout or atmospheric conditions (e.g. in L58) could be misleading in a control context where set-point may refer to command. A rather preferred term could be for instance "operating conditions".

   The term "operating set-point" has been changed to "operating condition" following the recommendation from the editor.

2. Eq. (4). The total farm power (P) could be named as ($P_{WF}$) to better match the subsequent reference in Section 4. The subscript WF could also be applied to Eq.(5).

   The total farm power has been named $P_{WF}$ in Eq. (4), Eq. (5), and in related occurences in the text.

3. L209, I'd suggest to include the explanation of the term multi-modality (L261: "the presence of multiple and distinct maxima") when the term is first mentioned in the text.

   An explanation of the term multi-modality has been added when first mentioned in the manuscript.

4. Table 2. Maybe you could include somehow that the second range for eps corresponds to the C1 constraint, since this is not explained until L454. Reference to Table 2 could also be made in L456.

   Table 2 has been modified with a clarification for *eps* values when referring to Constraint "C1". A reference to Table 2 has been added in Sect. 4.3, addressing the change in *eps* value.

5. Figure 2: The figure is not fully colour-blind friendly, specifically the lines corresponding to Gaussian and GCH models, which are very close, of similar line style and have the two most confused colours (green and red). Could you please kindly modify it to make it clearer for everyone?

   The line style of Figure 2 has been changed accordingly.

6. For the sake of clarity, please indicate in all figure captions to which layout corresponds the figure. Missing references in Fig. 2 (2x1 wind farm), Fig. 5-9 (5x5 wind farm) and Fig. 10-12 (Horns Rev wind farm).

   All figure captions now specify the corresponding wind farm layout.

7. For the sake of higher colour-blind friendliness, in the discussion about Fig. 5, when referring to the markers, please include not only the colour (blue, green, red), but also the type of the marker (square, triangle, circle). Please also include the type of marker in the figure caption. The same goes for the discussion of Fig. 9.

   Captions and discussions of Fig. 5 and Fig. 9 have been changed accordingly.

8. L295. Could you please provide any answer on RC1 comment about the statement "which saturates the upper bound constraints at 25° for all but the final turbine row", which refers the authors to previous results in the literature (Zong and Porté-Agel, 2020)? The comment is also related to those in Fig. 10 by RC1.

   In the manuscript, two wake models based on a Gaussian distribution of the streamwise velocity deficit are utilised. The first is the "Gaussian Bastankhah" wake model proposed by Bastankhah and Porté-Agel, 2014. The second is the "GCH" model developed by King et al., 2021. The wake model mentioned in RC1 refers to the "Gaussian Zong" model proposed by Zong and Porté-Agel, 2020, which also incorporates a Gaussian distribution for the streamwise velocity deficit. Both the "GCH" and "Gaussian Zong" models include secondary steering effects in their formulations, leading to optimal yaw settings exhibiting monotonically decreasing yaw angles with each turbine row. In contrast, the "Gaussian Bastankhah" model does not capture secondary steering effects, resulting in optimal yaw angles saturating the upper bound constraints at 25° for all but the final turbine row. Therefore, the discrepancies in optimal yaw settings between the "Gaussian Bastankhah" model used in our study and the "Gaussian Zong" model mentioned in RC1 are consistent with the respective modelling capabilities of these wake models.

9. L301. According to your answer to major concern 2 by RC1, you have selected the 10 random cases in Fig. 4 from the 50 cases studied in Section 4.3. It might be worth mentioning it in the section.

   A clarification has been added in Section 4.2.1.

10. L358. The comment refers to both Fig. 5-c and 5-g, so you could make reference to both sub-figures.

    The comment mentioned refers to the initial yaw angles shown in Fig. 5 (Fig. 5-c specifically for the Gaussian model although all models share the same initial conditions). The sentence has been rephrased to avoid this confusion.

11. L407. In the discussion about Figure 9, maybe the use of "the first case", "the second and third cases" is not the most adequate since this order does not match the order in the legend or in the velocity plots.

    The discussion of Figure 9 has been changed to avoid this confusion.

12. L412, I guess you refer to Figure 8-d specifically.

    Yes, this is now clarified in the manuscript.

13. L416, in the reference to the figure you could also include "Fig. 9-d"

    A reference to Fig. 9-d has been added.

14. L418, you could also include a reference to "Fig. 9-b"

    A reference to Fig. 9-b has been added.

15. L427, in the reference to Fig. 5, please specify to which sub-figure you want to make reference, if any in particular.

    The mentioned reference to Fig. 5 refers to all sub-figures. A clarification has been included in the text.

16. L454. When introducing constraint C1, could you please include an indication on why you choose the positive yaw angles as supposed to lead to more global maxima or to preferred solutions?

    A clarification has been added in Sect. 4.3 where constraint "C1" is first introduced.

17. L464. For the sake of clarity, please also include that the so-called nominal optimisation does not apply constraint C2 either, as RC1 suggested.

    A clearer explanation is now included.

18. L467, please make reference to Table 2.

    A reference to Table 2 has been added.

19. L476. To better complement the discussion and demonstrate the overall effect of the optimisation constraints, could you please include (e.g. in a table) the normalised farm power obtained by each of the optimisations depicted in Fig. 10? This might not be relevant in model comparisons for all the reasons discussed with referees, but can be of interest when analysing a particular model with different optimisation approaches. This may also help to differentiate the C1 and C1+C2 approaches. Do the results from the latter justify its greater complexity? Additionally, some comment about the number of runs required would be of interest, as RC1 commented.

    A new table (Table 3 in the revised manuscript) with the resulting farm power improvements from the optimisation cases depicted in Fig. 10 has been added with a related discussion in the text addressing the points of the above comment. Moreover, the number of optimisation runs required for the statistical analysis of the Horns Rev case is 50. This is specified in the fourth paragraph of Sect. 4.3 (L463 of the revised manuscript submitted in response to RC1 and RC2).

20. L504. Could you please better support the statement "For all wake models, the row-averaged probability density functions indicate an overall reduction in optimal yaw angle dependency to initial conditions for the constrained cases" by indicating from which specific aspect of the results this conclusion derives? (Comment by RC1).

    The increase in probability density function magnitudes for the constrained cases indicates a more consistent distribution of optimal yaw angles for different initialisations (i.e., improved robustness). The paragraph in the manuscript has been rephrased for clarity.

21. L506. Could you please specify in which subfigures it is observed the following statement: "Higher probability density function magnitudes can be observed, with more than double the values for some row distributions"? (Comment by RC1).

References to sub-figures have been added to the mentioned statement, as well as a clearer explanation.

22. There are several comments by RC1 pointing at the benefits that stochastic optimisation could also provide to the problem under study, referring to (Kuo et al., 2020). For the sake of completeness, the authors could include some comment (e.g. in the introduction) showing this as an alternative option, not covered in this manuscript but that could also be used to address the limitations of SLSQP approaches. It is true that the reference indicated by RC1 is now included in the introduction but just as part of a mere listing of optimisation techniques without further indication. If you do not consider RC1 comments as pertinent, please provide some explanation of why.

    Similarly to the algorithm employed in Kuo et al., 2020, TuRBO is a global stochastic optimisation method. Consequently, we believe stochastic optimisation is acknowledged in the manuscript as a valid choice to address the limitations associated with gradient-based algorithms. To provide clarity, an explicit statement highlighting the stochastic nature of TuRBO has been included in the introduction and in Sect. 3.1.

23. Some parts of the authors' answer to RC2 would be of interest in the text of the manuscript. For instance, aspects from the third paragraph of the answer to comment 1 could be nicely included in the discussion. Also, the main differences between SLSQP and TuRBO algorithms stated in your answer to comment 4.

    The suggested aspects have been included in the manuscript. In particular, a discussion on model robustness has been added to Sect. 4.2.1 (in lines 325–330 of the revised manuscript) and further insight into SLSQP and TuRBO comparison to Sect. 3.1 (in lines 195–197 of the revised manuscript).

24. Please check and harmonise the references to figures throughout the text so that they fulfil the journal guidelines: The abbreviation "Fig." should be used when it appears in running text and should be followed by a number unless it comes at the beginning of a sentence, e.g.: "The results are depicted in Fig. 5. Figure 9 reveals that...".

    References to figures have been adjusted to comply with the journal guidelines.

25. According to the journal guidelines, you will be requested to include a persistent identifier (DOI preferred) for all the references. So, you could already advance those modifications from now.

    A persistent identifier has been added to all references (DOI when available).

**Typos:**

1. Figure 5: required space "the(a)"

2. L508. Multizone and Gaussian nominal cases would be related to Fig. 12-a and 12-b instead of Fig. 12-b and 12-c.

3. L512. Fig. 12-c instead of Fig. 12-d?

4. L519. Please eliminate "a" in the sentence "This can achieved by a permuting the turbine".

5. L642. Subscript of the rotor velocity in the text would be with capital letter "R" to match the equations.

   We thank the editor for pointing these out. We have corrected all typos.

---

## Author Response (AR3)

**Authors' response to the editor's review of "Sensitivity analysis of wake steering optimisation for wind farm power maximisation"**

F. Gori, S. Laizet, A. Wynn

July, 2023

**Main comment**

The authors would like to thank the editor for the review of the paper. The authors' responses (in blue) to all comments (in black) are included in this document. Furthermore, following the detection and resolution of a software bug in the FLORIS framework, the related adjustments to the paper are also outlined below. All modifications are marked appropriately in the revised manuscript.

**Minor changes:**

1. L328, the statement "offers more physically accurate optimal yaw angles than the Gaussian model". The final accuracy of the model is out of the scope of the manuscript discussion (as already discussed with referees and indicated in the text, the paper is not about demonstrating which model is the best), and the statement is not supported by any external reference either. Maybe the sentence could be rephrased by considering that such model takes more physical effects into account, as stated in the previous paragraph in the manuscript, but without entering into accuracy discussions if you want to keep the stated scope.

   The statement has been rephrased to avoid the model comparison in terms of accuracy.

2. L530, when referring to the nominal cases, maybe you mean Fig. 12-a,b,c.

   Yes, thank you for spotting this. "d,e,f" has been replaced by "a,b,c".

**Changes due to software bug:**

The authors have identified and resolved a small bug in the published implementation of the FLORIS software version 2.4. The bug affected the GCH model, specifically the calculation of the effective yaw angles caused by secondary steering effects in negative yaw conditions. The FLORIS developer team has been notified (issue number 684), and the authors expect a resolution of the issue, which consists of a simple sign switch, from negative to positive, in the upcoming weeks.

In light of this, Figures 4 to 12 have been updated by incorporating the corrected results for the GCH model. All other presented results remain unaffected. Additionally, the references in the manuscript text regarding specific resulting values, such as farm power improvements, have been appropriately

updated. The main outcomes of the manuscript are the same. Please refer to the point below for more detailed information about the manuscript changes.

- Following the software correction, the farm power function for the GCH model exhibits high gradients rather than discontinuities. The discontinuities for this model were produced as a result of this bug. The authors found these discontinuities surprising and, in the reviewed submission, tried to provide an explanation to the reader. Such an explanation (which was provided in Figure 7) is no longer needed and has been removed. The authors have also revised the manuscript by replacing the term "discontinuities" with "high gradients" when referring to the GCH model. Note finally that the bug was only marginally affecting the power output of a wind turbine. The authors have re-run all experiments and have updated all modified values in the manuscript.

- The authors have updated Figure 5 with an initial condition "Test Case 1" that most faithfully highlights the optimal yaw angle dependency on initial yaw angles for the GCH model. The discussion of the presented results remains essentially unchanged. The initialisation condition is still shared among all models and is included in the statistical analysis of Section 4.2.3.

- As a result of the previous point, the planar slices through the objective function shown in Figure 6 exhibit slight variations due to the different optimal solutions for "Test Case 1". Furthermore, Figures 6-d and 6-h no longer show discontinuities for the GCH model.

- The authors have observed that negative signs in optimal yaw angles have a slightly greater influence on the initialisation sensitivity of the GCH model. While this behaviour is already mentioned and explained in the manuscript, the authors have further clarified it as one of the leading causes of GCH initialization sensitivity, along with column-wise sign switches in optimal yaw angles.

---

## Author Response (AR4)

**Authors' response to the editor's review of "Sensitivity analysis of wake steering optimisation for wind farm power maximisation"**

F. Gori, S. Laizet, A. Wynn

August, 2023

The authors would like to thank the editor for the review of the paper. The authors have made the necessary technical corrections by including "The convention for yaw misalignment involves a positive yaw angle signifying an anti-clockwise rotation of the nacelle when viewed from above." at L156-157 of the revised manuscript.